# Iterated INLA for State and Parameter Estimation in Nonlinear Dynamical Systems

**Rafael Anderka**[1]        **Marc Peter Deisenroth**[1]        **So Takao**[1,2]

[1]Centre for Artificial Intelligence, University College London, London, UK
[2]Department of Computing and Mathematical Sciences, California Institute of Technology, Pasadena, CA

## Abstract

Data assimilation (DA) methods use priors arising from differential equations to robustly interpolate and extrapolate data. Popular techniques such as ensemble methods that handle high-dimensional, nonlinear PDE priors focus mostly on state estimation, however can have difficulty learning the parameters accurately. On the other hand, machine learning based approaches can naturally learn the state and parameters, but their applicability can be limited, or produce uncertainties that are hard to interpret. Inspired by the Integrated Nested Laplace Approximation (INLA) method in spatial statistics, we propose an alternative approach to DA based on iteratively linearising the dynamical model. This produces a Gaussian Markov random field at each iteration, enabling one to use INLA to infer the state and parameters. Our approach can be used for arbitrary nonlinear systems, while retaining interpretability, and is furthermore demonstrated to outperform existing methods on the DA task. By providing a more nuanced approach to handling nonlinear PDE priors, our methodology offers improved accuracy and robustness in predictions, especially where data sparsity is prevalent.

## 1 INTRODUCTION

Physics-based modelling plays a major role in science and engineering even in today's landscape of machine learning, where heavy emphasis is placed on data-driven modelling. In applications, such as numerical weather prediction (NWP), the number of observations received daily, while plentiful, pales in comparison to the sheer dimensionality of the state—typically of the order of $\mathcal{O}(10^9)$, while the number of observations is of the order of $\mathcal{O}(10^7)$ Office [2024]. Under such data-scarce regimes, it is crucial to incorporate

expert knowledge into models so that we can extrapolate in regions outside of the data distribution (Figure 1), while equipping them with sound uncertainty estimates.

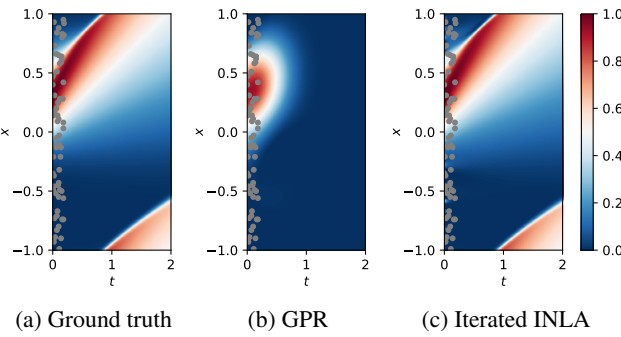

(a) Ground truth        (b) GPR        (c) Iterated INLA

Figure 1: Comparison of the predictions made from a non-physics-informed model (Gaussian process regression / GPR) vs. a physics-informed model (iterated INLA). The ground truth is a simulation of the 1D Burgers' equation. Gray dots are observation locations.

Data assimilation (DA) techniques have been devised to infer the state from data, when the model takes the form of a differential equation. In particular, when the dynamical model is linear Gaussian, one can solve the problem exactly using the Kalman filter/smoother; in nonlinear settings, one can compute approximate solutions via the extended Kalman filter/smoother or particle-based methods. However, in large-scale problems, such as NWP, these techniques are intractable due to their high computational and memory costs. This calls for further approximations, such as the ensemble Kalman filter/smoother or variational methods [Evensen et al., 2022]; however, they have their respective drawbacks. For example, ensemble Kalman methods make Gaussian approximations to non-Gaussian posteriors using a small number of particles, which can lead to noisy correlations and systematically small variances [Bannister, 2017]; variational algorithms, by themselves, do not even provide uncertainty estimates. Furthermore, NWP models often contain multiple parameters that are manually adjusted by the

modellers, which is an area that can be automated using machine learning [Schneider et al., 2017]. For example, ensemble Kalman methods can be extended to jointly infer the state and model parameters by augmenting the state vector (Evensen [2009], Bocquet and Sakov [2013]). However, this can only provide Gaussian approximations to the non-Gaussian joint posteriors, which can lead to instability. Further, it has been observed that ensemble Kalman methods struggle to accurately estimate parameters associated with stochastic terms in the model [DelSole and Yang, 2010].

Alternatively, there has been a recent surge of interest in integrating physical knowledge into machine learning (ML) models. Physics-informed neural networks (PINNs) introduced by Raissi et al. [2019] for example, use neural networks to parametrically represent the model state to estimate the state and parameters given data, by minimising a tailored loss function comprising a data fit term and a PDE residual term. Analogous ML-based methods using Gaussian processes (GPs) have also been proposed, such as AutoIP [Long et al., 2022], which replaces the neural network in PINNs by GPs to equip predictions with uncertainties using variational inference. A similar method has been proposed by Chen et al. [2021] that also provides convergence guarantees in the limit of increasing collocation points. However, all these methods encode the PDE knowledge artificially through the *likelihood*, instead of directly embedding them in the prior, which makes interpretability of their predicted uncertainties harder. On the other hand, latent force models [Alvarez et al., 2009], or the recent work by Nikitin et al. [2022] injects ODE/PDE knowledge directly into the GP prior by designing physics-informed kernels. However, their approach is limited to modelling linear PDEs, such as the heat and wave equations.

We overcome the limitations posed above by proposing a DA method inspired by the work Rue et al. [2013]. This is a statistical inference technique that represents GP priors by a Gaussian Markov Random Field (GMRF), enabling the posterior marginals on the state and model hyperparameters to be inferred efficiently using Integrated Nested Laplace Approximation (INLA) [Rue et al., 2009]. We extend their method to handle nonlinear PDE priors by iteratively linearising the PDE to produce GP approximations to the prior at each iteration, and subsequently using INLA to update the state and parameter estimates. In particular, this allows us to (1) use priors derived from arbitrary nonlinear PDEs without having to encode them in the likelihood, and (2) provide accurate non-Gaussian estimates to the state and posterior marginals, going beyond the setting of Gaussian approximations, necessarily imposed in variational inference or ensemble Kalman methods.

## 2 BACKGROUND

Physical systems are typically modelled by ordinary or partial differential equations. These are often augmented by stochastic noise terms to account for model uncertainty. In particular, for $0 < T < \infty$, consider a stochastic model over $t \in [0, T]$ of the form

$$\frac{\partial u}{\partial t}(x, t) = \mathcal{M}[u](x, t) + \dot{\mathcal{W}}(t, x), \quad x \in \mathring{D}, \quad (1)$$

$$\mathcal{B}u(x, t) = 0, \quad x \in \partial D, \quad (2)$$

$$u(x, 0) \sim \mathcal{N}(u_b, \mathcal{C}), \quad (3)$$

where (1) expresses the differential equation satisfied by $u(t, \cdot)$ in the interior of a compact domain $D$, perturbed by a spatio-temporal noise term $\dot{\mathcal{W}}$; (2) expresses the boundary conditions for a given operator $\mathcal{B}$ (e.g., the identity map or spatial derivatives), and (3) describes the initial condition, which we model as a Gaussian random element with mean $u_b$ and covariance $\mathcal{C}$. Denoting by $V$ a topological vector space where the solution $u(t, \cdot)$ to (1)–(3) lives for $t \in [0, T]$, we take $\mathcal{M}$ in (1) to be a possibly nonlinear operator on $V$ that may additionally depend on a set of parameters $\boldsymbol{\theta}$ that are a priori unknown. Equations (1)–(3) encode the prior knowledge we have about our state $u(t, x)$.

Consider observations $\boldsymbol{y}(t) \in \mathbb{R}^{d_y}$ of $u(t, x)$, modelled by the likelihood

$$p(\boldsymbol{y}_t | u_t) = \mathcal{N}(\boldsymbol{y}_t | \boldsymbol{h}_t(u_t), \sigma_y^2 \mathbf{I}), \quad (4)$$

where $\boldsymbol{h}_t : V \to \mathbb{R}^{d_y}$ is some observation operator at time $t$, which we assume to be linear. $\sigma_y$ is the standard deviation of the observation noise, which may be unknown and can be included in the set $\boldsymbol{\theta}$. At a high level, we are interested in computing an updated belief of the state $u(x, t)$ and parameters $\boldsymbol{\theta}$, given the observations $\{\boldsymbol{y}_t\}_{t \in [0, T]}$. Specifically, consider a numerical discretisation of (1)–(4), which we express in state-space form

$$\boldsymbol{u}_n = \boldsymbol{f}_{\boldsymbol{\theta}}(\boldsymbol{u}_{n-1}) + \boldsymbol{\epsilon}_n, \quad \boldsymbol{\epsilon}_n \sim \mathcal{N}(\mathbf{0}, \mathbf{Q}), \quad (5)$$

$$\boldsymbol{y}_n = \mathbf{H}_n \boldsymbol{u}_n + \boldsymbol{\eta}_n, \quad \boldsymbol{\eta}_n \sim \mathcal{N}(\mathbf{0}, \mathbf{R}), \quad (6)$$

for $n = 1, \ldots, N_t$ and $\boldsymbol{u}_0 \sim \mathcal{N}(\boldsymbol{u}_b, \mathbf{C})$. Here, $\boldsymbol{u}_b, \boldsymbol{u}_n \in \mathbb{R}^{d_u}$ are discretisations of the fields $u_b(x)$ and $u(t_n, \cdot)$ for $0 = t_0 < t_1 < \ldots < t_N = T$, the matrices $\mathbf{Q} \in \mathbb{R}^{d_u \times d_u}$, $\mathbf{R} \in \mathbb{R}^{d_y \times d_y}$ are the process and observation noise covariances, and $\boldsymbol{f}_{\boldsymbol{\theta}} : \mathbb{R}^{d_u} \to \mathbb{R}^{d_u}$, $\mathbf{H}_n \in \mathbb{R}^{d_y \times d_u}$ are the discretised dynamics and observation operators, respectively. Then letting $\boldsymbol{y} := \{\boldsymbol{y}_1, \ldots, \boldsymbol{y}_{N_t}\}$, we wish to compute

$$p(u_n^i | \boldsymbol{y}, \boldsymbol{\theta}) \propto \int p(\boldsymbol{y}_n | \boldsymbol{u}_n, \boldsymbol{\theta}) p(\boldsymbol{u}_n | \boldsymbol{\theta}) \mathrm{d}\boldsymbol{u}_n^{\backslash i} \quad (7)$$

for $i = 1, \ldots, d_u$ and $n = 1, \ldots, N_t$ if $\boldsymbol{\theta}$ is known, or

$$p(\boldsymbol{\theta} | \boldsymbol{y}) \propto \int p(\boldsymbol{y} | \boldsymbol{u}, \boldsymbol{\theta}) p(\boldsymbol{\theta}) \mathrm{d}\boldsymbol{u}, \quad (8)$$

$$p(u_n^i | \boldsymbol{y}) = \int p(u_n^i | \boldsymbol{y}, \boldsymbol{\theta}) p(\boldsymbol{\theta} | \boldsymbol{y}) \mathrm{d}\boldsymbol{\theta}, \quad (9)$$

where $\boldsymbol{u} := (\boldsymbol{u}_1, \ldots, \boldsymbol{u}_{N_t})$, if $\boldsymbol{\theta}$ is unknown. We refer to this as the data assimilation (DA) problem.

## 2.1 ENSEMBLE AND VARIATIONAL DA

In general, the distributions (7)–(9) can be approximated arbitrarily well using sequential Monte-Carlo (SMC) techniques [Chopin and Papaspiliopoulos, 2020]. However, this can be expensive to compute and moreover suffers from a so-called "weight-collapse" when $d_u \gg 1$, making it unreliable to use in high-dimensional settings [Bengtsson et al., 2008]. The ensemble Kalman filter/smoother (EnKF/S) [Evensen, 1994, Evensen and Van Leeuwen, 2000] has been proposed as an appealing alternative in high dimensions, which, like SMC, uses particles to empirically approximate the state distribution, but differs from it by only using information about the first two moments of the particles to condition on data. This effectively employs a Gaussian approximation to all distributions arising in the computations, making it more akin to the extended Kalman filter/smoother (ExKF/S). However, by taking the number of particles to be much smaller than $d_u$, computations in EnKF/S can be performed much more efficiently than in ExKF/S, enabling its use in high-dimensional problems such as weather forecasting. However, using a small number of particles can result in inaccurate uncertainty estimates [Bannister, 2017]. Moreover, when we jointly infer the parameters via state-augmentation [Evensen, 2009], parameters of the process noise cannot be inferred accurately [DelSole and Yang, 2010].

Variational methods, such as 4D-Var [Le Dimet and Talagrand, 1986], on the other hand reduce the Bayesian inference problem (7) to a MAP estimation problem, where one seeks to minimise a cost functional of the form (in this case, the *weak-constraint 4D-Var loss* $J = -\log p(\boldsymbol{u}|\boldsymbol{y}, \boldsymbol{\theta})$):

$$J[\boldsymbol{u}; \boldsymbol{\theta}] := \frac{1}{2} \sum_{n=1}^{N_t} \|\boldsymbol{y}_n - \mathbf{H}_n \boldsymbol{u}_n\|_{\mathbf{R}^{-1}}^2 \qquad (10)$$

$$+ \frac{1}{2} \sum_{n=1}^{N_t} \|\boldsymbol{u}_n - \boldsymbol{f}_{\boldsymbol{\theta}}(\boldsymbol{u}_{n-1})\|_{\mathbf{Q}^{-1}}^2 + \frac{1}{2} \|\boldsymbol{u}_0 - \boldsymbol{u}_b\|_{\mathbf{C}^{-1}}^2.$$

Here we used the shorthand $\|\boldsymbol{v}\|_{\mathbf{A}}^2 := \boldsymbol{v}^\top \mathbf{A} \boldsymbol{v}$. Optimising the cost (10), using e.g. a quasi-Newton method [Evensen et al., 2022], is tractable for high-dimensional problems. However, a shortcoming of the approach is that it does not directly provide any form of uncertainty estimates on $\boldsymbol{u}$.

## 2.2 INLA

In spatial statistics, the integrated nested Laplace approximation (INLA) [Rue et al., 2009] is a commonly used Bayesian inference method for latent Gaussian models. INLA consid-

ers a hierarchical Bayesian model of the form

$$\boldsymbol{\theta} \sim p_\Theta(\cdot), \qquad (11)$$

$$\boldsymbol{u}|\boldsymbol{\theta} \sim \mathcal{N}(\boldsymbol{\mu}_{\boldsymbol{u}}(\boldsymbol{\theta}), \mathbf{P}_{\boldsymbol{u}}^{-1}(\boldsymbol{\theta})) \qquad (12)$$

for some distribution $p_\Theta$ that is not necessarily Gaussian; $\boldsymbol{\mu}_{\boldsymbol{u}}(\boldsymbol{\theta}), \mathbf{P}_{\boldsymbol{u}}(\boldsymbol{\theta})$ are the mean vector and precision matrix of the latent process $\boldsymbol{u}$ conditioned on $\boldsymbol{\theta}$. Given the likelihood $p(\boldsymbol{y}|\boldsymbol{u}, \boldsymbol{\theta})$, INLA approximates the marginal posteriors $\{p(u_i|\boldsymbol{y})\}_{i=1}^{d_u}$ in (9) by numerical integration

$$p(u_i|\boldsymbol{y}) \approx \sum_{k=1}^{K} p(u_i|\boldsymbol{y}, \boldsymbol{\theta}_k) p(\boldsymbol{\theta}_k|\boldsymbol{y}) \Delta_k, \qquad (13)$$

where $\{\boldsymbol{\theta}_k\}_{k=1}^{K}$ are $K$ quadrature nodes for numerically integrating in $\boldsymbol{\theta}$-space, and $\{\Delta_k\}_{k=1}^{K}$ are volume elements in $\boldsymbol{\theta}$-space. When the likelihood is Gaussian, i.e., $p(\boldsymbol{y}|\boldsymbol{u}, \boldsymbol{\theta}) = \mathcal{N}(\boldsymbol{y}|\mathbf{H}\boldsymbol{u}, \mathbf{R})$ for some matrix $\mathbf{H}$ and noise covariance $\mathbf{R}$, then by standard computation, the posterior $p(\boldsymbol{u}|\boldsymbol{y}, \boldsymbol{\theta}) = \mathcal{N}(\boldsymbol{u}|\boldsymbol{\mu}_{\boldsymbol{u}|\boldsymbol{y}}(\boldsymbol{\theta}), \mathbf{P}_{\boldsymbol{u}|\boldsymbol{y}}^{-1}(\boldsymbol{\theta}))$ has the closed-form expression

$$\mathbf{P}_{\boldsymbol{u}|\boldsymbol{y}}(\boldsymbol{\theta}) = \mathbf{P}_{\boldsymbol{u}}(\boldsymbol{\theta}) + \mathbf{H}^\top \mathbf{R}^{-1} \mathbf{H} \qquad (14)$$

$$\boldsymbol{\mu}_{\boldsymbol{u}|\boldsymbol{y}}(\boldsymbol{\theta}) = \boldsymbol{\mu}_{\boldsymbol{u}}(\boldsymbol{\theta}) + \mathbf{P}_{\boldsymbol{u}|\boldsymbol{y}}(\boldsymbol{\theta})^{-1} \mathbf{H}^\top \mathbf{R}^{-1}(\boldsymbol{y} - \mathbf{H}\boldsymbol{\mu}_{\boldsymbol{u}}(\boldsymbol{\theta})). \qquad (15)$$

Provided that $\boldsymbol{u}$ is a GMRF, so that $\mathbf{P}_{\boldsymbol{u}}(\boldsymbol{\theta})$ is sparse, and assuming that $\mathbf{H}^\top \mathbf{R}^{-1} \mathbf{H}$ is also sparse, then the posterior mean (15) can be computed efficiently using a sparse Cholesky solver and the marginal posterior variances can be computed by Takahashi recursions [Takahashi, 1973, Rue and Martino, 2007] (see Appendix A.1 for details). For the marginal posterior on the parameters $p(\boldsymbol{\theta}|\boldsymbol{y})$, the following approximation is considered

$$\tilde{p}(\boldsymbol{\theta}|\boldsymbol{y}) \propto \left. \frac{p(\boldsymbol{u}, \boldsymbol{y}, \boldsymbol{\theta})}{\tilde{p}_G(\boldsymbol{u}|\boldsymbol{y}, \boldsymbol{\theta})} \right|_{\boldsymbol{u}=\boldsymbol{u}^*(\boldsymbol{\theta})}, \qquad (16)$$

where $\tilde{p}_G(\boldsymbol{u}|\boldsymbol{y}, \boldsymbol{\theta})$ is a Gaussian approximation to $p(\boldsymbol{u}|\boldsymbol{y}, \boldsymbol{\theta})$. In the Gaussian likelihood case, this is just $p(\boldsymbol{u}|\boldsymbol{y}, \boldsymbol{\theta})$. We also denoted $\boldsymbol{u}^*(\boldsymbol{\theta}) := \arg\max_{\boldsymbol{u}}[\log p(\boldsymbol{u}|\boldsymbol{y}, \boldsymbol{\theta})]$. Finally, the quadrature nodes $\boldsymbol{\theta}_k$ in (13) are selected from a regular grid in a transformed $\boldsymbol{\theta}$-space, such that it satisfies

$$|\log \tilde{p}(\boldsymbol{\theta}^*|\boldsymbol{y}) - \log \tilde{p}(\boldsymbol{\theta}_k|\boldsymbol{y})| < \delta, \qquad (17)$$

where $\boldsymbol{\theta}^* := \arg\max_{\boldsymbol{\theta}}[\log p(\boldsymbol{\theta}|\boldsymbol{y})]$ and for some acceptance threshold $\delta > 0$. We provide details of the selection criteria in Appendix A.3 and details for evaluating the expression (16) in Appendix A.2.

## 3 ITERATED INLA FOR NONLINEAR DA

While INLA is typically employed for latent fields that are modelled by GMRFs, here, we extend its applicability to particular non-Gaussian fields, namely, those generated by nonlinear SPDEs. By doing so, we obtain a new, principled method for jointly inferring the state and parameters in nonlinear dynamical systems.

## 3.1 LINEAR SETTING

Before considering the general setting of nonlinear SPDE priors, let us first consider the case when $\mathcal{M}$ in (1) is a linear differential operator. In this setting, one can build a GMRF representation of $u$ from this operator via the so-called *SPDE approach* by Lindgren et al. [2011]. To do this, we first discretise the differential operator

$$\mathcal{L} := \frac{\partial}{\partial t} - \mathcal{M}, \qquad (18)$$

using e.g., finite differences, which results in a sparse, banded matrix $\mathcal{L} \approx \mathbf{L} \in \mathbb{R}^{N \times N}$. Upon discretising with finite differences, the SPDE (1)–(3) can be approximated by a random matrix-vector system (see Appendix D)

$$\mathbf{L}\boldsymbol{u} = \boldsymbol{\xi}, \qquad (19)$$

where $\boldsymbol{\xi} \sim \mathcal{N}(0, \bar{\mathbf{Q}})$, for some positive definite matrix $\bar{\mathbf{Q}}$, which numerically represents the covariance structure of the space-time noise process $\dot{\mathcal{W}}$ in (1). For space-time white noise process, this simply reads $\bar{\mathbf{Q}} = \frac{\sigma_u^2}{\Delta t \Delta x}\mathbf{I}$, where $\sigma_u > 0$ is the spectral density of the noise, which can be treated as another unknown parameter in the set $\boldsymbol{\theta}$ (see Appendix D for the derivation). If the matrix $\mathbf{L}^\top \bar{\mathbf{Q}}^{-1}\mathbf{L}$ is invertible, then we deduce from (19) that

$$\boldsymbol{u} \sim \mathcal{N}(\mathbf{0}, (\mathbf{L}^\top \bar{\mathbf{Q}}^{-1}\mathbf{L})^{-1}), \qquad (20)$$

which is a GMRF if the prior precision $\mathbf{P}_{\boldsymbol{u}} := \mathbf{L}^\top \bar{\mathbf{Q}}^{-1}\mathbf{L}$ is sparse. Given observations $\boldsymbol{y}$ of $\boldsymbol{u}$, we directly apply INLA (Section 2.2) to infer the marginal posteriors $p(u_i|\boldsymbol{y})$ of the state, and if the model $\mathcal{M}$ also contains some unknown parameters $\boldsymbol{\theta}$, then its marginal posteriors $p(\theta_j|\boldsymbol{y})$ as well.

## 3.2 NONLINEAR SETTING

In the nonlinear setting, we aim to follow a similar strategy by constructing a GMRF from the model (1)–(3) and using INLA to jointly estimate the state $u$ and parameter $\boldsymbol{\theta}$ from the data $\boldsymbol{y}$. However, the nonlinearity of the operator $\mathcal{M}$ leads to non-Gaussianity of $u$, preventing us from directly obtaining a GMRF representation of $p(u)$ by discretisation, as we saw in Section 3.1. To overcome this, we adopt an iterative strategy, whereby at each iteration $n$, we consider a Gaussian approximation to $p(u)$ by linearising the model $\mathcal{M}$ around a point $u_0^{(n)}$. That is,

$$\mathcal{M}[u] \approx \mathcal{M}[u_0^{(n)}] + \mathcal{M}_0^{(n)}(u - u_0^{(n)}) \qquad (21)$$

$$= (\mathcal{M}[u_0^{(n)}] - \mathcal{M}_0^{(n)}u_0^{(n)}) + \mathcal{M}_0^{(n)}u \qquad (22)$$

for some linear operator $\mathcal{M}_0^{(n)}$. Then, the spatio-temporal operator $\mathcal{L}$ in (18) can be approximated by an affine operator

$$\mathcal{L}[u] \approx \mathcal{L}_0^{(n)}u - r_0^{(n)}, \qquad (23)$$

where

$$\mathcal{L}_0^{(n)}u := \frac{\partial u}{\partial t} - \mathcal{M}_0^{(n)}u, \quad \text{and} \qquad (24)$$

$$r_0^{(n)} := \mathcal{M}[u_0^{(n)}] - \mathcal{M}_0^{(n)}u_0^{(n)} \qquad (25)$$

$$= \mathcal{L}_0^{(n)}u_0^{(n)} - \mathcal{L}[u_0^{(n)}]. \qquad (26)$$

Now, considering a finite-difference discretisation in space-time, denote by $\boldsymbol{u}, \boldsymbol{r}^{(n)}$ the corresponding vector representation of the fields $u, r_0^{(n)}$, and by $\mathbf{L}^{(n)}$ the corresponding matrix representation of the linear operator $\mathcal{L}_0^{(n)}$. By (23), this gives us the following approximation to system (1)–(3)

$$\mathbf{L}^{(n)}\boldsymbol{u} = \boldsymbol{r}^{(n)} + \boldsymbol{\xi}, \qquad (27)$$

where $\boldsymbol{\xi} \sim \mathcal{N}(\mathbf{0}, \bar{\mathbf{Q}})$ is the discretised noise process. Hence, as in the linear setting, we find the following Gaussian approximation to $p(\boldsymbol{u})$ at the $n$-th iteration:

$$\tilde{p}_G^{(n)}(\boldsymbol{u}) = \mathcal{N}(\boldsymbol{u} \,|\, (\mathbf{L}^{(n)})^{-1}\boldsymbol{r}^{(n)}, (\mathbf{L}^{(n)\top}\bar{\mathbf{Q}}^{-1}\mathbf{L}^{(n)})^{-1}). \qquad (28)$$

This is a GMRF, provided that the approximate prior precision $\mathbf{P}_{\boldsymbol{u}}^{(n)} := \mathbf{L}^{(n)\top}\bar{\mathbf{Q}}^{-1}\mathbf{L}^{(n)}$ is sparse. In practice, we compute the mean in (28) as $(\mathbf{L}^{(n)\top}\mathbf{L}^{(n)})^{-1}\mathbf{L}^{(n)\top}\boldsymbol{r}^{(n)}$, which we found to be more numerically stable. We can further compute the corresponding posterior $\tilde{p}_G^{(n)}(\boldsymbol{u}|\boldsymbol{y},\boldsymbol{\theta})$ using (14)–(15). With this, we are in position to apply INLA.

To summarise, our iterated INLA methodology entails (i) linearising our model $\mathcal{L}$ around a point $u_0^{(n)}$, (ii) obtain an approximate GMRF representation of the state $u$ using (28), (iii) apply INLA on this model to compute an estimate for the marginal posteriors $p(u_i|\boldsymbol{y})$ and $p(\theta_j|\boldsymbol{y})$, and (iv) iterate steps (i)–(iii) with an updated linearisation point $u_0^{(n+1)}$. In the following, we address this last point regarding how to update the linearisation point, depending on whether we know the model parameters or not.

*Remark* 3.1. We note that our method is similar to the iterative INLA method in the `inlabru` R package [Lindgren and Suen, 2023] for handling nonlinear predictors in GLMs. The main difference is where the nonlinearity appears - In `inlabru`, this arises by directly taking nonlinear transformations to a GMRF prior, whereas in our setting the nonlinearity is inherent in the SPDE defining the prior. This subtle difference leads to different formalisms.

### 3.2.1 Known parameters

In the case where the model parameters $\boldsymbol{\theta}$ are known, we choose to take the resulting posterior mean

$$\boldsymbol{\mu}_{\boldsymbol{u}|\boldsymbol{y}}^{(n)}(\boldsymbol{\theta}) := \mathbb{E}_{\tilde{p}_G^{(n)}(\boldsymbol{u}|\boldsymbol{y},\boldsymbol{\theta})}[\boldsymbol{u}] \qquad (29)$$

**Algorithm 1** Iterated INLA with known parameters

1: **Input:** observations $\boldsymbol{y}$, parameters $\boldsymbol{\theta}$, damping coeff. $\gamma$
2: **Initialise:** $u_0^{(0)}$, $n = 0$
3: **while** $\boldsymbol{u}_0^{(n)}$ has not converged **do**
4:     $\mathcal{L}_0^{(n)} \leftarrow$ Linearise operator $\mathcal{L}$ around $u^{(n)}$
5:     $r_0^{(n)} \leftarrow$ Compute residual $\mathcal{L}_0^{(n)} u^{(n)} - \mathcal{L}[u^{(n)}]$
6:     $\mathbf{L}^{(n)}, \boldsymbol{r}^{(n)} \leftarrow$ Discretise $\mathcal{L}_0^{(n)}$ and $r_0^{(n)}$
7:     $\mathbf{P}_{\boldsymbol{u}}^{(n)}(\boldsymbol{\theta}) \leftarrow \mathbf{L}^{(n)\top} \bar{\mathbf{Q}}^{-1} \mathbf{L}^{(n)}$    (Prior precision)
8:     $\boldsymbol{\mu}_{\boldsymbol{u}}^{(n)}(\boldsymbol{\theta}) \leftarrow (\mathbf{L}^{(n)})^{-1} \boldsymbol{r}^{(n)}$    (Prior mean)
9:     $\mathbf{P}_{\boldsymbol{u}|\boldsymbol{y}}^{(n)}(\boldsymbol{\theta}) \leftarrow$ Equation (14)    (Posterior precision)
10:    $\boldsymbol{\mu}_{\boldsymbol{u}|\boldsymbol{y}}^{(n)}(\boldsymbol{\theta}) \leftarrow$ Equation (15)   (Posterior mean)
11:    $\boldsymbol{u}_0^{(n+1)} \leftarrow (1 - \gamma) \boldsymbol{u}_0^{(n)} + \gamma \boldsymbol{\mu}_{\boldsymbol{u}|\boldsymbol{y}}^{(n)}(\boldsymbol{\theta})$
12:    $n \leftarrow n + 1$
13: **end while**
14: $\boldsymbol{v}_{\boldsymbol{u}|\boldsymbol{y}}^{(\infty)}(\boldsymbol{\theta}) \leftarrow$ Takahashi recursion on $\mathbf{P}_{\boldsymbol{u}|\boldsymbol{y}}^{(\infty)}(\boldsymbol{\theta})$
15: **return** $\boldsymbol{\mu}_{\boldsymbol{u}|\boldsymbol{y}}^{(\infty)}(\boldsymbol{\theta}), \boldsymbol{v}_{\boldsymbol{u}|\boldsymbol{y}}^{(\infty)}(\boldsymbol{\theta})$

---

computed using (15) with prior (28), as the next linearisation point $\boldsymbol{u}_0^{(n+1)}$ in the iteration. In practice, we perform a damped update of the form

$$\boldsymbol{u}_0^{(n+1)} = (1 - \gamma) \boldsymbol{u}_0^{(n)} + \gamma \boldsymbol{\mu}_{\boldsymbol{u}|\boldsymbol{y}}^{(n)}(\boldsymbol{\theta}) \qquad (30)$$

to aid stability, where the parameter $\gamma \in (0, 1]$ is a tunable damping coefficient. At convergence ($n = \infty$), we further compute the marginal posterior variances

$$\boldsymbol{v}_{\boldsymbol{u}|\boldsymbol{y}}^{(\infty)}(\boldsymbol{\theta}) := \mathtt{diag}\left( \mathbf{P}_{\boldsymbol{u}|\boldsymbol{y}}^{(\infty)}(\boldsymbol{\theta})^{-1} \right) \qquad (31)$$

using Takahashi recursion [Rue and Martino, 2007]; this computation only has to be performed once at the end and not at every iteration. We summarise the full process in Algorithm 1.

Below, we show that updating $\boldsymbol{u}_0^{(n)}$ according to (30) is a sound choice, as it is identical to using the Gauss–Newton method to optimise the weak-constraint 4D-Var cost (10).

**Proposition 3.2.** *The damped update of the linearisation point $\boldsymbol{u}_0^{(n)}$ in (30) is equivalent to minimising the weak-constraint 4D-Var cost (10) using Gauss–Newton.*

*Proof.* Appendix B.1.     □

This implies firstly, that we have guaranteed convergence of our algorithm in the same setting where the Gauss–Newton method converges (e.g. Theorem 10.1 in Nocedal and Wright [1999]), and secondly, we can interpret the output of Algorithm 1 as the marginals of an *approximate* Laplace approximation to $p(\boldsymbol{u}|\boldsymbol{y}, \boldsymbol{\theta})$ that is close to the true Laplace approximation when the model is weakly nonlinear (see Appendix B.3).

### 3.2.2 Unknown parameters

In the case where the parameters $\boldsymbol{\theta}$ are unknown, we adopt the INLA methodology to jointly infer the state and parameters of the system as follows: Once obtaining the approximate Gaussian posterior $\tilde{p}_G^{(n)}(\boldsymbol{u}|\boldsymbol{y}, \boldsymbol{\theta})$, we compute an approximation of the marginal posterior $p(\boldsymbol{\theta}|\boldsymbol{y})$ by

$$\tilde{p}^{(n)}(\boldsymbol{\theta}|\boldsymbol{y}) \propto \left. \frac{p(\boldsymbol{u}, \boldsymbol{y}, \boldsymbol{\theta})}{\tilde{p}_G^{(n)}(\boldsymbol{u}|\boldsymbol{y}, \boldsymbol{\theta})} \right|_{\boldsymbol{u}=\boldsymbol{\mu}_{\boldsymbol{u}|\boldsymbol{y}}^{(n)}(\boldsymbol{\theta})} . \qquad (32)$$

The marginal posteriors $\{p(u_i|\boldsymbol{y})\}_{i=1}^{d_u}$ can then be approximated by numerical integration

$$\tilde{p}^{(n)}(u_i|\boldsymbol{y}) = \sum_k \tilde{p}_G^{(n)}(u_i|\boldsymbol{y}, \boldsymbol{\theta}_k^{(n)}) \tilde{p}^{(n)}(\boldsymbol{\theta}_k^{(n)}|\boldsymbol{y}) \Delta_k, \quad (33)$$

where the selection of the quadrature nodes $\boldsymbol{\theta}_k^{(n)}$ and volume elements $\Delta_k$ follow in the same way as vanilla INLA. Note that neither the approximate parameter estimate (32) nor the state estimate (33) are Gaussians (however, the latter is a mixture of Gaussians).

Looking at (33), it is natural to consider the following update rule to obtain the next linearisation point

$$\bar{\boldsymbol{u}}^{(n)} := \sum_k \boldsymbol{\mu}_{\boldsymbol{u}|\boldsymbol{y}}^{(n)}(\boldsymbol{\theta}_k^{(n)}) \tilde{p}^{(n)}(\boldsymbol{\theta}_k^{(n)}|\boldsymbol{y}) \Delta_k \qquad (34)$$

$$\boldsymbol{u}_0^{(n+1)} = (1 - \gamma) \boldsymbol{u}_0^{(n)} + \gamma \bar{\boldsymbol{u}}^{(n)}, \qquad (35)$$

for some $\gamma \in (0, 1]$ and $\boldsymbol{\mu}_{\boldsymbol{u}|\boldsymbol{y}}^{(n)}(\boldsymbol{\theta})$ is defined in (29). Let us call this the type-I update rule. We also consider another approach, where the parameter-averaging in (34) instead takes place on the natural parameters of $\tilde{p}_G^{(n)}(\boldsymbol{u}|\boldsymbol{y}, \boldsymbol{\theta})$, i.e.,

$$\bar{\mathbf{P}}^{(n)} := \sum_k \mathbf{P}_{\boldsymbol{u}|\boldsymbol{y}}^{(n)}(\boldsymbol{\theta}_k^{(n)}) \tilde{p}^{(n)}(\boldsymbol{\theta}_k^{(n)}|\boldsymbol{y}) \Delta_k, \qquad (36)$$

$$\bar{\boldsymbol{b}}^{(n)} := \sum_k \mathbf{P}_{\boldsymbol{u}|\boldsymbol{y}}^{(n)}(\boldsymbol{\theta}_k^{(n)}) \boldsymbol{\mu}_{\boldsymbol{u}|\boldsymbol{y}}^{(n)}(\boldsymbol{\theta}_k^{(n)}) \tilde{p}^{(n)}(\boldsymbol{\theta}_k^{(n)}|\boldsymbol{y}) \Delta_k, \quad (37)$$

$$\bar{\boldsymbol{u}}^{(n)} := (\bar{\mathbf{P}}^{(n)})^{-1} \bar{\boldsymbol{b}}^{(n)}, \qquad (38)$$

$$\boldsymbol{u}_0^{(n+1)} = (1 - \gamma) \boldsymbol{u}_0^{(n)} + \gamma \bar{\boldsymbol{u}}^{(n)}. \qquad (39)$$

We call this the type-II update rule. Using the type-II updates, we obtain a result analogous to Proposition 3.2 in the unknown parameter setting, where instead, the sequence $\{\boldsymbol{u}_0^{(n)}\}$ progressively minimises a "parameter-averaged" 4D-Var cost. We state this below.

**Proposition 3.3.** *Updating the linearisation point $\boldsymbol{u}_0^{(n)}$ according to (36)–(39) is an approximate Gauss–Newton method for minimising the parameter-averaged 4D-Var cost $\mathbb{E}_{p(\boldsymbol{\theta}|\boldsymbol{y})}[-\log p(\boldsymbol{u}|\boldsymbol{y}, \boldsymbol{\theta})]$.*

*Proof.* Appendix B.2.     □

**Algorithm 2** Iterated INLA with unknown parameters

1: **Input:** observations $\boldsymbol{y}$, damping coefficient $\gamma$
2: **Initialise:** $u_0^{(0)}$, $n = 0$
3: **while** $\boldsymbol{u}_0^{(n)}$ has not converged **do**
4: $\quad \mathcal{L}_0^{(n)} \leftarrow$ Linearise operator $\mathcal{L}$ around $u_0^{(n)}$
5: $\quad r_0^{(n)} \leftarrow \mathcal{L}_0^{(n)} u_0^{(n)} - \mathcal{L}[u_0^{(n)}]$
6: $\quad \mathbf{L}^{(n)}, \boldsymbol{r}^{(n)} \leftarrow$ Discretise $\mathcal{L}_0^{(n)}$ and $r_0^{(n)}$
7: $\quad \tilde{p}_G^{(n)}(\boldsymbol{u}|\boldsymbol{\theta}) \leftarrow \mathcal{N}(\boldsymbol{u}|(\mathbf{L}^{(n)})^{-1}\boldsymbol{r}^{(n)}, (\mathbf{L}^{(n)\top}\bar{\mathbf{Q}}^{-1}\mathbf{L}^{(n)})^{-1})$
8: $\quad \tilde{p}_G^{(n)}(\boldsymbol{u}|\boldsymbol{y},\boldsymbol{\theta}) \leftarrow$ Equations (14)–(15)
9: $\quad \tilde{p}^{(n)}(\boldsymbol{\theta}|\boldsymbol{y}) \leftarrow$ Equation (32)
10: $\quad$ Obtain quadrature nodes $\{\boldsymbol{\theta}_k^{(n)}\}_k$ satisfying (17)
11: $\quad \bar{\boldsymbol{u}} \leftarrow$ Equation (34) for type-I or (36)–(38) for type-II
12: $\quad \boldsymbol{u}_0^{(n+1)} \leftarrow (1-\gamma)\boldsymbol{u}_0^{(n)} + \gamma\bar{\boldsymbol{u}}$
13: $\quad n \leftarrow n+1$
14: **end while**
15: Compute state marginals at convergence:
16: **for** $i = 1, \ldots, d_u$ **do**
17: $\quad \tilde{p}_G^{(\infty)}(u_i|\boldsymbol{y},\boldsymbol{\theta}_k^{(\infty)}) \leftarrow$ Takahashi recursion
18: $\quad \tilde{p}^{(\infty)}(u_i|\boldsymbol{y}) \leftarrow \sum_k \tilde{p}_G^{(\infty)}(u_i|\boldsymbol{y},\boldsymbol{\theta}_k^{(\infty)})\tilde{p}^{(\infty)}(\boldsymbol{\theta}_k^{(\infty)}|\boldsymbol{y})\Delta_k$
19: **end for**
20: **return** $\left\{\tilde{p}^{(\infty)}(u_i|\boldsymbol{y})\right\}_{i=1}^{d_u}, \left\{\tilde{p}^{(\infty)}(\boldsymbol{\theta}|\boldsymbol{y})\right\}_{j=1}^{|\boldsymbol{\theta}|}$

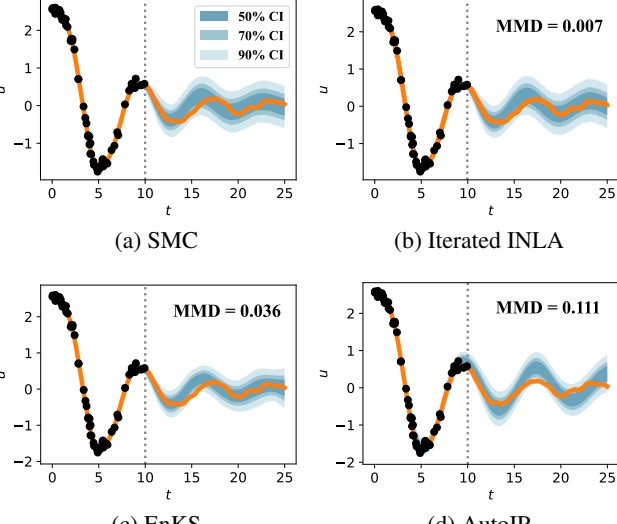

Figure 2: Comparison of the marginal state estimates $p(u_i|\boldsymbol{y})$ on the pendulum experiment. We display the credible intervals (CI) in blue shades; black dots are noisy observations from a sample simulation, displayed in orange. For methods (b)–(d), we display the maximum mean discrepancy (MMD) from the SMC result (a), which we take as the gold standard. Iterated INLA performs best both qualitatively and in terms of the MMD score.

By Jensen's inequality, we have that

$$-\log p(\boldsymbol{u}|\boldsymbol{y}) \leq \mathbb{E}_{p(\boldsymbol{\theta}|\boldsymbol{y})}[-\log p(\boldsymbol{u}|\boldsymbol{y},\boldsymbol{\theta})]. \qquad (40)$$

Thus, the minima of the parameter-averaged 4DVar cost can be seen as approximating the mode of $p(\boldsymbol{u}|\boldsymbol{y})$; the converged value $\boldsymbol{u}_0^{(\infty)}$ using the type-II update is therefore interpreted as an approximate MAP estimator for $\boldsymbol{u}$ given $\boldsymbol{y}$.

Using either update rules, we obtain uncertainty estimates on the predictions using the Gaussian mixture (33) at $n = \infty$, where the variances of the Gaussians $\tilde{p}_G^{(\infty)}(u_i|\boldsymbol{y},\boldsymbol{\theta}_k^{(\infty)})$ are computed using the Takahashi recursion. It is unnecessary to compute (33) at every iteration but only at the end. We summarise the full process in Algorithm 2.

*Remark* 3.4. The computational cost of iterated INLA is $\mathcal{O}(N_i I)$, where $N_i$ is the number of iterations and $I$ is the complexity of one interation of INLA (see Section A.1 for more details). There are no significant differences in the costs between type I and II updates. In general, this is cheaper than running particle MC, which requires a large number of particles to accurately estimate the state and parameter posteriors. However, it is more costly than running EnKS, which only scales linearly in the number of time steps and cubically in the ensemble size – the latter is typically chosen to be small.

## 4  EXPERIMENTS

In this section, we evaluate the ability of iterated INLA to infer the state and parameters on several benchmark nonlinear dynamical systems. In the first part, we consider inference on a low dimensional nonlinear SDE, where the goal is to compare against a "gold standard" SMC method. In the second part, we benchmark on several spatio-temporal nonlinear PDE systems to test the robustness of our method in the noise-free setting and compare the results against different baselines. Details can be found in Appendix C.

### 4.1  STOCHASTIC NONLINEAR PENDULUM

The goal of this experiment is to evaluate the accuracy of iterated INLA for inferring the state and parameters on a low dimensional system. We compare the results against a sequential Monte Carlo (SMC) baseline, which recovers the distributions $p(\boldsymbol{\theta}|\boldsymbol{y})$ and $p(\boldsymbol{u}|\boldsymbol{y})$ accurately as we are in a low dimensional setting. We therefore use these as "ground truths" that one can compare against. For the dynamics model, we consider the stochastic pendulum system

$$\frac{\mathrm{d}^2 u}{\mathrm{d}t^2} + b\frac{\mathrm{d}u}{\mathrm{d}t} + c\sin u = \sigma_u \dot{W}_t, \qquad (41)$$

with unknown parameters $b, c$ and $\sigma_u$. Our aim is to infer these alongside the state $u$ from noisy observations $\boldsymbol{y}$ of a

sample trajectory of (41). The observation noise amplitude $\sigma_y$ is also taken to be unknown and is to be inferred too. The precise details on the experimental set up can be found in Appendix C.2.

As baselines, we considered vanilla RBF-GP regression (GPR), the ensemble Kalman smoother (EnKS) and AutoIP [Long et al., 2022]. For EnKS, we use the state-augmentation approach [Evensen, 2009] to jointly infer the state and model parameters $(b, c, \sigma_u)$. We also consider an iterative extension of EnKS (iEnKS) proposed in Bocquet and Sakov [2013], which can be used for joint state and paramter estimation. However, these methods do not accommodate learning of the observation noise $\sigma_y$, so we fix this to the ground truth value in the EnKS / iEnKS experiments. AutoIP is capable of learning all four parameters, however it can only learn point estimates by gradient descent. Therefore, we initialise them with fixed values, set to the mode of the respective priors. For GPR, we only learn the hyperparameters of the RBF kernel by type-II maximum likelihood estimation [Rasmussen and Williams, 2006].

|        | RMSE            | MNLL             | MMD             |
|--------|-----------------|------------------|-----------------|
| GPR    | $0.26 \pm 0.03$ | $-0.08 \pm 0.03$ | $0.59 \pm 0.17$ |
| EnKS   | $0.18 \pm 0.01$ | $-0.50 \pm 0.08$ | $0.29 \pm 0.10$ |
| iEnKS  | $0.21 \pm 0.02$ | $1.02 \pm 0.74$  | $0.74 \pm 0.21$ |
| AutoIP | $\mathbf{0.14 \pm 0.02}$ | $-0.11 \pm 0.24$ | $0.58 \pm 0.16$ |
| iINLA-I  | $0.23 \pm 0.06$ | $-0.52 \pm 0.12$ | $0.29 \pm 0.16$ |
| iINLA-II | $0.18 \pm 0.01$ | $\mathbf{-0.67 \pm 0.06}$ | $\mathbf{0.17 \pm 0.06}$ |

Table 1: State prediction accuracy (RMSE+MNLL) and MMD from the SMC baseline on the pendulum experiment. We display the mean and standard errors across ten seeds.

We display the results across ten random simulations of (41) in Table 1. We compare the root mean square error (RMSE) and the mean negative log-likelihood (MNLL) of the estimated marginal state posteriors $\tilde{p}(u_i|\boldsymbol{y})$. The RMSE was computed using the appropriate estimators for $\boldsymbol{u}$ for each model—for GPR, EnKS and AutoIP, we took the predictive means; for iterated INLA, we took the converged linearisation points $\boldsymbol{u}_0^{(\infty)}$. In addition, we compared the maximum mean discrepancy (MMD) [Gretton et al., 2012] of the estimates $\tilde{p}(u_i|\boldsymbol{y})$ from $p(u_i|\boldsymbol{y})$, computed using SMC. The MMD measures how close two distributions are based on samples from the respective distributions. We also compare both update rules for iterated INLA, which we abbreviate as iINLA-I and II respectively. Table 1 shows that, while AutoIP shows the best performance on the RMSE, it performs poorly on the MNLL, likely due to overconfident predictions. On the other hand, both iINLA methods outperform the other models on the MNLL, suggesting a good calibration of the uncertainties. Using the type-II update, iINLA is also shown to have the closest results to SMC, as indicated by the low MMD score. Interestingly, the results obtained

by the type-II update outperforms those obtained by type-I across all metrics. While it is difficult to understand exactly why this occurs, it is possible that the fact that linearisation occurs around the MAP estimate of $p(\boldsymbol{u}|\boldsymbol{y})$ using type-II updates (Proposition 3.3) helps to improve the performance.

In Figure 2, we plot the state uncertainties produced by SMC, AutoIP, EnKS and iINLA-II on a single random seed. Here, we can see that the uncertainties generated by iINLA-II is nearly identical to the SMC output. This is also reflected in the lower MMD score. We also display the estimates of the parameters in Figure 3, where we plot a heatmap of the estimated distribution on the parameters $b$ and $\sigma_u$, computed using (a) iINLA-II, and (b) EnKS. As a reference, we also display the marginals on the parameters $b, \sigma_u$ computed using SMC in blue. We see that both methods achieve similar results to SMC for estimating $b$. However, for $\sigma_u$, we see that while the estimates from iINLA-II agree closely with the results from SMC, the estimate from EnKS is significantly different. This behaviour is consistent with previous observations that ensemble methods struggle to learn parameters associated with stochastic terms in the equation [DelSole and Yang, 2010]. Iterated INLA in contrast can get accurate estimates on the stochastic parameters. As a reference, to produce the SMC results in Figure 3 took $\approx 25$ minutes on an M1 Macbook Pro, whereas it took $\approx 1$ minute to produce analogous results using iINLA (Table 2).

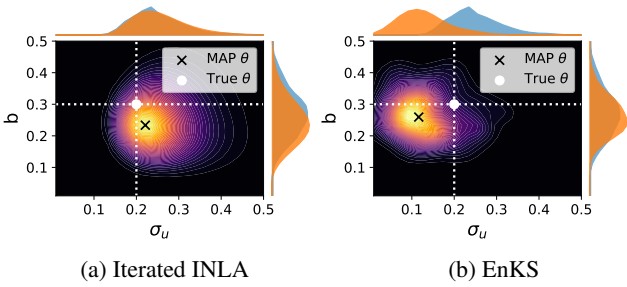

(a) Iterated INLA       (b) EnKS

Figure 3: Estimated marginal posterior densities for the $b$ parameter and the system noise parameter $\sigma_u$ using (a) iINLA-II and (b) EnKS for the pendulum experiment. The marginal distributions are displayed in orange on the respective axes. We also plot the marginal distributions obtained by SMC in blue. For the $\sigma_u$ parameter, the estimates obtained by EnKS diverges from SMC, while iterated INLA recovers it correctly.

## 4.2 PDE BENCHMARKS

In this experiment, we evaluate the performance of iINLA on several benchmark spatio-temporal PDE datasets, including the Burgers' equation, Allen-Cahn (AC) equation and the Korteweg-de Vries (KdV) equation. Details of these systems and its linearisations can be found in Appendices C.4–C.5. For each PDE, we generated a deterministic tra-

| Method | Run time (s) |
|---|---|
| SMC (10,000 samples) | $1541 \pm 38$ |
| iINLA-II (25 iterations) | $67.26 \pm 0.94$ |
| EnKS (100 ensembles) | $3.07 \pm 0.31$ |

Table 2: Comparison of run times between SMC, iterated INLA (type II) and EnKS to produce the parameter estimates in Figure 3. We display the mean and standard deviation of run times across five different runs on an M1 Macbook Pro.

jectory representing the ground truth. Then we randomly sampled noisy observations from the generated fields, which we used as a training set to recover the original field and the parameters used to generate them. We selected one parameter to learn per model. However for iINLA, we additionally need to train the process noise parameter $\sigma_u > 0$, whose real value is zero. It is therefore also of interest to see how iINLA performs under this mismatched model scenario.

We compared the performance of iINLA against the same baselines of GPR, EnKS, iEnKS and AutoIP. Their results are summarised in Table 3. Generally, we find that iINLA and EnKS perform better than the other models on both metrics (with the exception of the Burgers' experiment, where iEnKS performs marginally better). AutoIP tends to produce over-smoothed results and fails to learn the correct parameter, leading to an MNLL that is even worse than GPR's. The differences between type I and II updates in iINLA were negligible here.

For the Burgers' experiment, the performance of EnKS, iEnKS and iINLA are similar, with the iEnKS slightly ourperforming the others on both the RMSE and the MNLL. However, we encountered numerical stability issues with the EnKS and iEnKS using a fourth-order Runge–Kutta scheme with a timestep of $\Delta t = 0.02$ when jointly learning the state and parameters (we did not encounter this issue when learning just the state). Hence, this required us to use a more sophisticated solver in Kassam and Trefethen [2005] with an order of magnitude smaller timestep of $\Delta t = 10^{-3}$ to run the simulations reliably. Iterated INLA did not have this issue and ran reliably at the original timestep, using a basic central difference scheme for discretisation. On the Allen-Cahn example, we see that EnKS outperforms iINLA on both metrics (iEnKS performed significantly worse on this example). To understand this, it helps to see that the uncertainties generated by iINLA is generally higher than those generated by EnKF (Figure 4). This is due to the existence of the small but positive process noise $\sigma_u$ that cannot be removed from iINLA. Upon training, this converged to $10^{-3}$. Hence, even in regions where predictions can be more confident, the uncertainty cannot go below this value, leading to slighly smoother and underconfident predictions (note that the uncertainty in EnKS goes down to $10^{-5}$). In the KdV example, we instead see that iINLA performs bet-

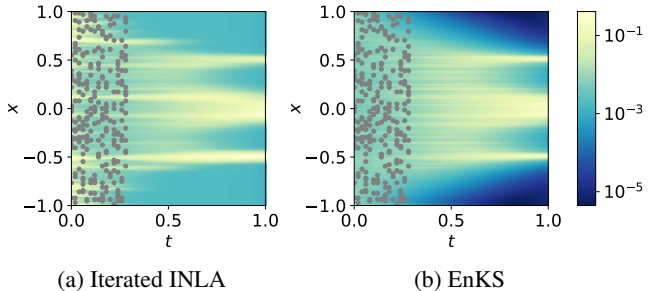

(a) Iterated INLA  (b) EnKS

Figure 4: Comparison of the predicted standard deviations on the Allen-Cahn example. The predictions are generally underconfident for iINLA due to the presence of $\sigma_u > 0$. Gray dots are observation locations.

ter than EnKS. Again, we encountered numerical stability issues with EnKS on the KdV example and is likely the cause for the poor performance of EnKS. On the other hand, we found that iINLA is numerically robust and converges consistently, without the need for grid upsampling. We plot the outputs of all methods for each PDE in Appendix E.

In terms of parameter estimation, we find that iINLA recovers the correct values for all three PDEs reliably, as shown in Table 4. This is despite our initial guesses (the prior modes) being reasonably far from the true values. Here, $\nu$, $C$ and $\lambda_1$ refer to the trainable parameters in the Burgers', Allen-Cahn and KdV models respectively.

## 5 DISCUSSION AND CONCLUSION

In this paper, we proposed an algorithm based on the INLA methodology that effectively learns the state and the parameters in nonlinear dynamical systems without resorting to expensive MCMC. This is achieved by iteratively linearising the dynamical model, where one can apply INLA to infer the state and parameters. We prove that this is approximately identical to the Gauss-Newton method for minimising the 4D-Var loss, and demonstrate experimentally that it is numerically robust; it also produces accurate non-Gaussian estimates of the latent variables. Issues remain regarding the scalability of the method: INLA is typically employed for moderately-sized problems in two or three physical dimensions. It is difficult to see this being used in very large-scale applications, such as numerical weather forecasting. We also do not consider non-Gaussian likelihoods here, although this should be a straightforward extension by adopting nested Laplace approximations. We also have not exploited the Markovian structure in the temporal component à la filtering/smoothing, which may help to speed up the algorithm. While the results are promising for the toy models considered here, further investigation is necessary to determine how our method fares in realistic medium-scale scenarios such as optical tomography Arridge and Schotland [2009] and nuclear fusion control [Morishita et al., 2024].

| | Burgers' | | Allen-Cahn | | Korteweg-de Vries | |
|---|---|---|---|---|---|---|
| | RMSE | MNLL | RMSE | MNLL | RMSE | MNLL |
| GPR | $0.119 \pm 0.004$ | $-0.959 \pm 0.028$ | $0.468 \pm 0.003$ | $-0.283 \pm 0.021$ | $0.461 \pm 0.008$ | $0.521 \pm 0.018$ |
| EnKS | $0.008 \pm 0.001$ | $-3.67 \pm 0.11$ | $\mathbf{0.028 \pm 0.001}$ | $\mathbf{-4.08 \pm 0.076}$ | $0.228 \pm 0.029$ | $-0.010 \pm 0.263$ |
| iEnKS | $\mathbf{0.006 \pm 0.001}$ | $\mathbf{-3.97 \pm 0.05}$ | $0.062 \pm 0.002$ | $11.67 \pm 1.18$ | $0.131 \pm 0.021$ | $2807 \pm 650$ |
| AutoIP | $0.018 \pm 0.003$ | $17.2 \pm 10.2$ | $0.389 \pm 0.008$ | $16.2 \pm 4.2$ | $0.270 \pm 0.007$ | $0.677 \pm 0.067$ |
| iINLA-I | $0.009 \pm 0.001$ | $-3.49 \pm 0.39$ | $0.053 \pm 0.003$ | $-2.30 \pm 0.60$ | $\mathbf{0.010 \pm 0.000}$ | $\mathbf{-3.28 \pm 0.03}$ |
| iINLA-II | $0.009 \pm 0.001$ | $-3.49 \pm 0.34$ | $0.053 \pm 0.004$ | $-2.95 \pm 0.14$ | $\mathbf{0.010 \pm 0.000}$ | $\mathbf{-3.28 \pm 0.04}$ |

Table 3: Performance of iINLA and baseline models on three PDE benchmarks. We display the mean and the standard error of the RMSE and MNLL across five different seeds for each system, where the randomness is due to observation sampling.

| Parameters | $\nu$ | $C$ | $\lambda_1$ |
|---|---|---|---|
| True values | 0.02 | 5.0 | 1.0 |
| Prior modes | 0.05 | 3.0 | 0.5 |
| Estimates | $0.023 \pm 0.001$ | $5.07 \pm 0.07$ | $0.996 \pm 0.004$ |

Table 4: Estimated parameter values using iINLA. We display the mean and standard error of the estimated values (i.e. posterior modes) across five different seeds.

## Code Availability

The code accompanying this paper is available at `https://github.com/rafaelanderka/iter-inla`.

## Author Contributions

Conceptualisation: ST; Methodology: RA, ST; Software: RA; Writing - original draft: ST; Writing - Review and Editing: RA, MPD; Supervision: MPD, ST. All authors approved the final submitted draft.

## Acknowledgements

ST is supported by a Department of Defense Vannevar Bush Faculty Fellowship held by Prof. Andrew Stuart, and by the SciAI Center, funded by the Office of Naval Research (ONR), under Grant Number N00014-23-1-2729.

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

# Iterated INLA for State and Parameter Estimation in Nonlinear Dynamical Systems
# (Supplementary Material)

**Rafael Anderka**[1]                **Marc Peter Deisenroth**[1]                **So Takao**[1,2]

[1]Centre for Artificial Intelligence, University College London, London, UK
[2]Department of Computing and Mathematical Sciences, California Institute of Technology, Pasadena, CA

## A   INLA DETAILS

In this appendix, we provide further details on the INLA algorithm used to infer the state and parameter marginal posterior estimates for prior models defined by GMRFs.

### A.1   SPARSE LINEAR SOLVE AND MATRIX INVERSION

A key component of INLA is to exploit the sparsity of the GMRF precisions to accelerate posterior inference. In particular, computing the posterior mean and variance (see (14)–(15)) requires taking large matrix inversions, which, if performed naively using dense matrices, scales as $\mathcal{O}(d_u^3)$. This is too expensive for most spatial or spatio-temporal modelling purposes. Fortunately, algorithms exist to speed up these computations significantly when the matrices are sparse. For solving linear systems (e.g. (15)), the matrix to invert (i.e., the posterior precision) is symmetric and positive definite. Hence, it is appropriate to use a Cholesky solver here. A sparse Cholesky solver is available through the `scikit-sparse` library, which provide Python bindings to the CHOLMOD C library [Chen et al., 2008]. The latter provides fast routines for sparse Cholesky factorisation among other things. By using the sparse Cholesky decomposition, one is able to reduce the initial $\mathcal{O}(d_u^3)$ complexity of solving the linear problem to $\mathcal{O}(d_u)$, $\mathcal{O}(d_u^{1/2})$, $\mathcal{O}(d_u^2)$ for problems in one, two and three-physical dimensions, respectively

For our purposes, we also need to recover marginal variances from the precision matrix, which in theory requires a matrix inversion – again not feasible in our setting. To overcome this, INLA employs the so-called *Takahashi recursion* to recover the marginal variances from the Cholesky factors of the precision (see [Rue and Martino, 2007, Section 2] for the full algorithm). Once the Cholesky factors are available, the typical cost for Takahashi recursion is $\mathcal{O}(d_u(\log d_u)^2)$. While an implementation of the Takahashi recursion is available in R through the R-INLA package Lindgren and Rue [2015], no suitable Python package was available. We therefore extended the `scikit-sparse` library to include existing routines for fast Takahashi recursions implemented in C, ensuring compatibility with the pre-existing framework. We hope this contribution will be incorporated into the main branch of the library, thereby allowing easy access to fast Takahashi recursion in Python for other researchers, and extending the contributions of this work further.

### A.2   COMPUTING $\tilde{p}(\boldsymbol{\theta}|\boldsymbol{y})$

We recall that INLA computes the marginal state posteriors by numerical integration

$$p(u_i|\boldsymbol{y}) = \int p(u^i|\boldsymbol{y}, \boldsymbol{\theta}) p(\boldsymbol{\theta}|\boldsymbol{y}) \mathrm{d}\boldsymbol{\theta} \approx \sum_{k=1}^{K} \tilde{p}(u_i|\boldsymbol{y}, \boldsymbol{\theta}_k) \tilde{p}(\boldsymbol{\theta}_k|\boldsymbol{y}) \Delta_k, \tag{42}$$

where $\tilde{p}(u_i|\boldsymbol{y}, \boldsymbol{\theta})$ and $\tilde{p}(\boldsymbol{\theta}|\boldsymbol{y})$ are approximations to the distributions $p(u_i|\boldsymbol{y}, \boldsymbol{\theta})$ and $p(\boldsymbol{\theta}|\boldsymbol{y})$. When the likelihood is Gaussian, then we can compute the posterior $p(u_i|\boldsymbol{y}, \boldsymbol{\theta})$ exactly and efficiently using the techniques in Appendix A.1. Hence, we don't

require further approximations, i.e., we can take $\tilde{p}(u_i|\boldsymbol{y}, \boldsymbol{\theta}) = p(u_i|\boldsymbol{y}, \boldsymbol{\theta})$. For $p(\boldsymbol{\theta}|\boldsymbol{y})$, we use the approximation

$$\tilde{p}(\boldsymbol{\theta}|\boldsymbol{y}) \propto \left.\frac{p(\boldsymbol{u}, \boldsymbol{y}, \boldsymbol{\theta})}{p(\boldsymbol{u}|\boldsymbol{y}, \boldsymbol{\theta})}\right|_{\boldsymbol{u}=\boldsymbol{\mu}_{\boldsymbol{u}|\boldsymbol{y}}(\boldsymbol{\theta})}, \tag{43}$$

which can be understood as a Laplace approximation of $p(\boldsymbol{\theta}|\boldsymbol{y})$ in the sense of Tierney and Kadane [1986]. To compute this explicitly, we consider its log-transform

$$\log \tilde{p}(\boldsymbol{\theta}|\boldsymbol{y}) = [\log p(\boldsymbol{u}, \boldsymbol{y}, \boldsymbol{\theta}) - \log p(\boldsymbol{u}|\boldsymbol{y}, \boldsymbol{\theta})]_{\boldsymbol{u}=\boldsymbol{\mu}_{\boldsymbol{u}|\boldsymbol{y}}(\boldsymbol{\theta})} + \text{const.} \tag{44}$$

$$= \left[\sum_{i=1}^{|\boldsymbol{\theta}|} \log p(\theta_i) + \log p(\boldsymbol{u}|\boldsymbol{\theta}) + \log p(\boldsymbol{y}|\boldsymbol{u}, \boldsymbol{\theta}) - \log p(\boldsymbol{u}|\boldsymbol{y}, \boldsymbol{\theta})\right]_{\boldsymbol{u}=\boldsymbol{\mu}_{\boldsymbol{u}|\boldsymbol{y}}(\boldsymbol{\theta})} + \text{const.} \tag{45}$$

$$\begin{aligned}
= &\sum_{i=1}^{|\boldsymbol{\theta}|} \log p(\theta_i) + \frac{1}{2}\log|\mathbf{Q}_{\boldsymbol{u}}(\boldsymbol{\theta})| - \frac{1}{2}(\boldsymbol{\mu}_{\boldsymbol{u}|\boldsymbol{y}}(\boldsymbol{\theta}) - \boldsymbol{\mu}_{\boldsymbol{u}}(\boldsymbol{\theta}))^\top \mathbf{Q}_{\boldsymbol{u}}(\boldsymbol{\theta})(\boldsymbol{\mu}_{\boldsymbol{u}|\boldsymbol{y}}(\boldsymbol{\theta}) - \boldsymbol{\mu}_{\boldsymbol{u}}(\boldsymbol{\theta})) - \frac{M}{2}\log 2\pi \\
&+ \frac{1}{2}\log|\mathbf{R}^{-1}| - \frac{1}{2}(\boldsymbol{y} - \mathbf{H}\boldsymbol{\mu}_{\boldsymbol{u}|\boldsymbol{y}}(\boldsymbol{\theta}))^\top \mathbf{R}^{-1}(\boldsymbol{y} - \mathbf{H}\boldsymbol{\mu}_{\boldsymbol{u}|\boldsymbol{y}}(\boldsymbol{\theta})) - \frac{N}{2}\log 2\pi \\
&+ \frac{1}{2}\log|\mathbf{Q}_{\boldsymbol{u}|\boldsymbol{y}}(\boldsymbol{\theta})| - \frac{M}{2}\log 2\pi + \text{const.},
\end{aligned} \tag{46}$$

which can be evaluated numerically (ignoring the constant, whose value we don't know). Then we take its exponential to get $\tilde{p}(\boldsymbol{\theta}|\boldsymbol{y})$ up to a constant. Regarding this constant, we can absorb it implicitly into the area element $\Delta_k$ in the expression (42). This is achieved by relying on the identity

$$1 = \int_{-\infty}^{\infty} p(u_i|\boldsymbol{y})\mathrm{d}u_i \overset{(42)}{\approx} \sum_{k=1}^{K}\left(\int_{-\infty}^{\infty}\tilde{p}(u_i|\boldsymbol{y}, \boldsymbol{\theta}_k)\mathrm{d}u_i\right)\tilde{p}(\boldsymbol{\theta}_k|\boldsymbol{y})\Delta_k = \sum_{k=1}^{K}\tilde{p}(\boldsymbol{\theta}_k|\boldsymbol{y})\Delta_k. \tag{47}$$

Assuming that $\Delta_k = \Delta$ for all $k = 1, \ldots, K$ and replacing $\tilde{p}(\boldsymbol{\theta}_k|\boldsymbol{y})$ by its unnormalised counterpart $f(\boldsymbol{\theta}_k|\boldsymbol{y}) := Z\tilde{p}(\boldsymbol{\theta}_k|\boldsymbol{y})$ for $Z := \int f(\boldsymbol{\theta}_k|\boldsymbol{y})\mathrm{d}\boldsymbol{\theta}$, we find

$$\tilde{\Delta} := \Delta/Z = \frac{1}{\sum_{k=1}^{K} f(\boldsymbol{\theta}_k|\boldsymbol{y})}. \tag{48}$$

Thus, we have

$$(42) = \sum_{k=1}^{K} p(u_i|\boldsymbol{y}, \boldsymbol{\theta}_k)f(\boldsymbol{\theta}_k|\boldsymbol{y})\tilde{\Delta}, \tag{49}$$

which does not require knowledge of the normalisation constant $Z$. Next, we discuss how to select the quadrature nodes $\{\boldsymbol{\theta}_k\}_{k=1}^{K}$ in the above expression.

## A.3 SELECTION OF THE QUADRATURE NODES

In INLA, the quadrature nodes $\boldsymbol{\theta}_k$ in (42) are selected according to the following steps.

**Step 1.** Locate the mode $\boldsymbol{\theta}_*$ of $\tilde{p}(\boldsymbol{\theta}|\boldsymbol{y})$ by numerically optimising its log-transform $\log \tilde{p}(\boldsymbol{\theta}|\boldsymbol{y})$ as given above. This typically requires a quasi-Newton method to circumvent computing the Hessian directly. Here, the gradient, if unavailable, can be approximated via finite-difference methods and second derivatives are constructed using the difference between successive gradient vectors [Rue et al., 2009]. We can also use derivative-free search, such as the Nelder-Mead method, which does not require computation of the gradient. We adopt the latter in our experiments, available in `scipy`'s `optimize` module.

**Step 2.** Compute the Hessian matrix $\mathbf{H} := \nabla^2 \log \tilde{p}(\boldsymbol{\theta}|\boldsymbol{y})|_{\boldsymbol{\theta}=\boldsymbol{\theta}_*}$ at the mode $\boldsymbol{\theta}_*$ using finite differences (FD). Note that the inverse of this Hessian $\mathbf{H}^{-1}$ is exactly equal to the covariance matrix of a Gaussian approximation of $\tilde{p}(\boldsymbol{\theta}|\boldsymbol{y})$, as $\mathbf{H}$ captures

the curvature around its mode. We then compute the eigendecomposition of $\mathbf{H}^{-1} = \mathbf{V}\boldsymbol{\Lambda}\mathbf{V}^{\top}$ to identify the principal axes along which to explore $\tilde{p}(\boldsymbol{\theta}|\boldsymbol{y})$ for efficiency. This allows us to use the reparametrisation

$$\boldsymbol{\theta}(\boldsymbol{z}) = \boldsymbol{\theta}_* + \mathbf{V}\boldsymbol{\Lambda}^{\frac{1}{2}}\boldsymbol{z}, \tag{50}$$

which ensures we correct for rotation and scale of $\tilde{p}(\boldsymbol{\theta}|\boldsymbol{y})$.

**Step 3.** Generate samples of $\log \tilde{p}(\boldsymbol{\theta}|\boldsymbol{y})$ that cover the bulk of its probability mass, using the above parametrisation for $\boldsymbol{\theta}$. Specificallly, the original INLA paper proposes that to find the bulk of the mass of $\tilde{p}(\boldsymbol{\theta}|\boldsymbol{y})$, we can sample regularly spaced points $\{\boldsymbol{\theta}_k\}_{k=1}^K$ in $\boldsymbol{z}$-space, and combinations of these points as long as they fulfill that

$$|\log \tilde{p}(\boldsymbol{\theta}_k|\boldsymbol{y}) - \log \tilde{p}(\boldsymbol{\theta}_*|\boldsymbol{y})| < \delta \tag{51}$$

Here, $\delta > 0$ is a threshold that can be tuned to balance accuracy and efficiency. These samples of $\log \tilde{p}(\boldsymbol{\theta}|\boldsymbol{y})$ will be used for numerical integration to find marginals such $\tilde{p}(u_i|\boldsymbol{y})$.

We refer the readers to the original manuscript Rue et al. [2009] for more details.

# B  PROOFS AND DISCUSSIONS OF RESULTS

In this appendix, we provide further details on the results Proposition 3.2 and Proposition 3.3 regarding the connection of iterated INLA with (weak-constraint) 4D-Var data assimilation. We provide proofs and discuss implications of the results. For ease of presentation, we first rewrite the weak-constraint 4D-Var cost (10) in the following form:

$$J[\boldsymbol{u}] = \frac{1}{2}(\boldsymbol{y} - \bar{\mathbf{H}}\boldsymbol{u})^{\top}\bar{\mathbf{R}}^{-1}(\boldsymbol{y} - \bar{\mathbf{H}}\boldsymbol{u}) + \frac{1}{2}\mathcal{L}[\boldsymbol{u}]^{\top}\bar{\mathbf{Q}}^{-1}\mathcal{L}[\boldsymbol{u}]. \tag{52}$$

Here, we denoted $\boldsymbol{y} = (\boldsymbol{y}_1, \ldots, \boldsymbol{y}_{N_t})^{\top}$, $\boldsymbol{u} = (\boldsymbol{u}_0, \boldsymbol{u}_1, \ldots, \boldsymbol{u}_{N_t})^{\top}$,

$$\bar{\mathbf{R}} := \mathtt{diag}(\underbrace{\mathbf{R}, \ldots, \mathbf{R}}_{N_t \text{ times}}), \quad \bar{\mathbf{Q}} := \mathtt{diag}(\mathbf{C}, \underbrace{\mathbf{Q}, \ldots, \mathbf{Q}}_{N_t \text{ times}}), \tag{53}$$

$$\bar{\mathbf{H}} := \left(\mathbf{0}, \quad \mathtt{diag}(\mathbf{H}_1, \ldots, \mathbf{H}_{N_t})\right), \tag{54}$$

and $\mathcal{L}[\boldsymbol{u}]$ is a vector in $\mathbb{R}^{d_u(N_t+1)}$ of the form $\mathcal{L}[\boldsymbol{u}] = (\boldsymbol{\ell}_0, \ldots, \boldsymbol{\ell}_{N_t})$, where

$$\boldsymbol{\ell}_i = \begin{cases} \boldsymbol{u}_i - \boldsymbol{f}_{\boldsymbol{\theta}}(\boldsymbol{u}_{i-1}), & \text{if} \quad i = 1, \ldots, N_t, \\ \boldsymbol{u}_i - \boldsymbol{u}_b, & \text{if} \quad i = 0 \end{cases} \in \mathbb{R}^{d_u}. \tag{55}$$

If there are missing observations at certain times, say $t_n$, then we just set $\boldsymbol{y}_n \equiv \mathbf{0}$ and $\mathbf{H}_n \equiv \mathbf{0}$.

## B.1  PROOF OF PROPOSITION 3.2

**Proposition 3.2.** *The damped update of the linearisation point $\boldsymbol{u}_0^{(n)}$ in (30) is equivalent to minimising the weak-constraint 4D-Var cost (10) using Gauss–Newton.*

*Proof.* The Gauss–Newton iteration for minimising (52) reads

$$\boldsymbol{u}_0^{(n+1)} = \boldsymbol{u}_0^{(n)} - \gamma \mathbf{B}^{-1}\nabla J[\boldsymbol{u}_0^{(n)}] \tag{56}$$

$$= \boldsymbol{u}_0^{(n)} - \gamma \mathbf{B}^{-1}\left(\mathbf{L}^{(n)\top}\bar{\mathbf{Q}}^{-1}\mathcal{L}[\boldsymbol{u}_0^{(n)}] + \bar{\mathbf{H}}^{\top}\bar{\mathbf{R}}^{-1}(\mathbf{H}\boldsymbol{u}_0^{(n)} - \boldsymbol{y})\right) \tag{57}$$

where $\gamma \in (0, 1)$ is the learning rate, $\mathbf{L}^{(n)} := \nabla\mathcal{L}[\boldsymbol{u}_0^{(n)}]$ and

$$\mathbf{B} := \bar{\mathbf{H}}^{\top}\bar{\mathbf{R}}^{-1}\bar{\mathbf{H}} + \mathbf{L}^{(n)\top}\bar{\mathbf{Q}}^{-1}\mathbf{L}^{(n)} \tag{58}$$

is the preconditioner, given by the Gauss–Newton approximation to the Hessian of $J$. Next, denoting

$$m^{(n)} := \mathbf{L}^{(n)} u_0^{(n)} - \mathcal{L}[u_0^{(n)}], \tag{59}$$

we manipulate the above expression for the Gauss–Newton iteration as follows

$$(56) = u_0^{(n)} - \gamma \mathbf{B}^{-1} \Big[ (\mathbf{B} u_0^{(n)} - \mathbf{B} u_0^{(n)}) + \mathbf{L}^{(n)\top} \bar{\mathbf{Q}}^{-1} \mathcal{L}[u_0^{(n)}] + \bar{\mathbf{H}}^\top \bar{\mathbf{R}}^{-1} (\bar{\mathbf{H}} u_0^{(n)} - y) \Big] \tag{60}$$

$$= (1-\gamma) u_0^{(n)} + \gamma \mathbf{B}^{-1} \Big[ \mathbf{B} u_0^{(n)} - \mathbf{L}^{(n)\top} \bar{\mathbf{Q}}^{-1} \mathcal{L}[u_0^{(n)}] - \bar{\mathbf{H}}^\top \bar{\mathbf{R}}^{-1} (\bar{\mathbf{H}} u_0^{(n)} - y) \Big] \tag{61}$$

$$= (1-\gamma) u_0^{(n)} + \gamma \mathbf{B}^{-1} \Big[ \left( \bar{\mathbf{H}}^\top \bar{\mathbf{R}}^{-1} \bar{\mathbf{H}} + \mathbf{L}^{(n)\top} \bar{\mathbf{Q}}^{-1} \mathbf{L}^{(n)} \right) u_0^{(n)} - \mathbf{L}^{(n)\top} \bar{\mathbf{Q}}^{-1} \mathcal{L}[u_0^{(n)}] - \bar{\mathbf{H}}^\top \bar{\mathbf{R}}^{-1} (\bar{\mathbf{H}} u_0^{(n)} - y) \Big] \tag{62}$$

$$= (1-\gamma) u_0^{(n)} + \gamma \mathbf{B}^{-1} \Big[ \mathbf{L}^{(n)\top} \bar{\mathbf{Q}}^{-1} \left( \mathbf{L}^{(n)} u_0^{(n)} - \mathcal{L}[u_0^{(n)}] \right) + \bar{\mathbf{H}}^\top \bar{\mathbf{R}}^{-1} y \Big] \tag{63}$$

$$= (1-\gamma) u_0^{(n)} + \gamma \mathbf{B}^{-1} \Big[ \mathbf{L}^{(n)\top} \bar{\mathbf{Q}}^{-1} m^{(n)} + \bar{\mathbf{H}}^\top \bar{\mathbf{R}}^{-1} y \Big]. \tag{64}$$

We claim that

$$\mu_{u|y}^{(n)}(\boldsymbol{\theta}) := \mathbb{E}_{\tilde{p}_G^{(n)}(u|y,\boldsymbol{\theta})}[u] = \mathbf{B}^{-1} \Big[ \mathbf{L}^{(n)\top} \bar{\mathbf{Q}}^{-1} m^{(n)} + \bar{\mathbf{H}}^\top \bar{\mathbf{R}}^{-1} y \Big], \tag{65}$$

which implies that (64) is indeed the expression for the damped update of the state estimate. To see this, we recall that

$$\tilde{p}_G^{(n)}(u|y,\boldsymbol{\theta}) \propto \tilde{p}_G^{(n)}(u|\boldsymbol{\theta}) p(y|u,\boldsymbol{\theta}), \tag{66}$$

where

$$\tilde{p}_G^{(n)}(u|\boldsymbol{\theta}) = \mathcal{N}(u \,|\, (\mathbf{L}^{(n)})^{-1} m^{(n)}, (\mathbf{L}^{(n)\top} \bar{\mathbf{Q}} \, \mathbf{L}^{(n)})^{-1}) \tag{67}$$

is the approximate prior at the $n$-th iteration, and the likelihood reads

$$p(y|u,\boldsymbol{\theta}) = \mathcal{N}(y \,|\, \bar{\mathbf{H}} u, \bar{\mathbf{R}}). \tag{68}$$

Then, a standard computation for Gaussians shows that

$$\tilde{p}_G^{(n)}(u|y,\boldsymbol{\theta}) = \mathcal{N}(u|\mu_{u|y}^{(n)}(\boldsymbol{\theta}), \mathbf{P}_{u|y}^{(n)}(\boldsymbol{\theta})^{-1}), \tag{69}$$

where

$$\mathbf{P}_{u|y}^{(n)}(\boldsymbol{\theta}) = \bar{\mathbf{H}}^\top \bar{\mathbf{R}}^{-1} \bar{\mathbf{H}} + \mathbf{L}^{(n)\top} \bar{\mathbf{Q}}^{-1} \mathbf{L}^{(n)} = \mathbf{B}, \quad \text{and} \tag{70}$$

$$\mu_{u|y}^{(n)}(\boldsymbol{\theta}) = \mathbf{P}_{u|y}^{(n)}(\boldsymbol{\theta})^{-1} \Big[ \mathbf{L}^{(n)\top} \bar{\mathbf{Q}}^{-1} m^{(n)} + \bar{\mathbf{H}}^\top \bar{\mathbf{R}}^{-1} y \Big]. \tag{71}$$

In particular, this shows that (65) holds, proving our claim. □

## B.2  PROOF OF PROPOSITION 3.3

**Proposition 3.3.** *Updating the linearisation point $u_0^{(n)}$ according to equations (36)–(39) is an approximate Gauss–Newton method for minimising the parameter-averaged 4D-Var cost $\mathbb{E}_{p(\boldsymbol{\theta}|y)}[-\log p(u|y,\boldsymbol{\theta})]$.*

*Proof.* Let $J_{\boldsymbol{\theta}}$ be the 4DVar cost (52), with the dependence on $\boldsymbol{\theta}$ made explicit. Then we have

$$\mathbb{E}_{p(\boldsymbol{\theta}|y)}\big[ -\log p(u|y,\boldsymbol{\theta}) \big] = \mathbb{E}_{p(\boldsymbol{\theta}|y)}\big[ J_{\boldsymbol{\theta}}[u] \big] \tag{72}$$

$$= \frac{1}{2} \mathbb{E}_{p(\boldsymbol{\theta}|y)} \Big[ (y - \bar{\mathbf{H}}_{\boldsymbol{\theta}} u)^\top \bar{\mathbf{R}}_{\boldsymbol{\theta}}^{-1} (y - \bar{\mathbf{H}}_{\boldsymbol{\theta}} u) + \frac{1}{2} \mathcal{L}_{\boldsymbol{\theta}}[u]^\top \bar{\mathbf{Q}}_{\boldsymbol{\theta}}^{-1} \mathcal{L}_{\boldsymbol{\theta}}[u] \Big]. \tag{73}$$

The Gauss–Newton iteration for minimising the cost (73) reads

$$u_0^{(n+1)} = u_0^{(n)} - \gamma \mathbf{B}^{-1} \nabla_u \mathbb{E}_{p(\boldsymbol{\theta}|y)}\big[ J_{\boldsymbol{\theta}}[u_0^{(n)}] \big] \tag{74}$$

$$= u_0^{(n)} - \gamma \mathbf{B}^{-1} \mathbb{E}_{p(\boldsymbol{\theta}|y)}\big[ \nabla_u J_{\boldsymbol{\theta}}[u_0^{(n)}] \big] \tag{75}$$

$$= u_0^{(n)} - \gamma \mathbf{B}^{-1} \mathbb{E}_{p(\boldsymbol{\theta}|y)} \Big[ \mathbf{L}_{\boldsymbol{\theta}}^{(n)\top} \bar{\mathbf{Q}}_{\boldsymbol{\theta}}^{-1} \mathcal{L}_{\boldsymbol{\theta}}[u_0^{(n)}] + \bar{\mathbf{H}}_{\boldsymbol{\theta}}^\top \bar{\mathbf{R}}_{\boldsymbol{\theta}}^{-1} (\bar{\mathbf{H}}_{\boldsymbol{\theta}} u_0^{(n)} - y) \Big], \tag{76}$$

where $\gamma \in (0, 1)$ is the learning rate, $\mathbf{L}_{\boldsymbol{\theta}}^{(n)} := \nabla\mathcal{L}_{\boldsymbol{\theta}}[\boldsymbol{u}_0^{(n)}]$ and

$$\mathbf{B} := \mathbb{E}_{p(\boldsymbol{\theta}|\boldsymbol{y})}\big[\bar{\mathbf{H}}_{\boldsymbol{\theta}}^{\top}\bar{\mathbf{R}}_{\boldsymbol{\theta}}^{-1}\bar{\mathbf{H}}_{\boldsymbol{\theta}} + \mathbf{L}_{\boldsymbol{\theta}}^{(n)\top}\bar{\mathbf{Q}}_{\boldsymbol{\theta}}^{-1}\mathbf{L}_{\boldsymbol{\theta}}^{(n)}\big] \tag{77}$$

is the preconditioner, given by the Gauss–Newton approximation of the Hessian of $\mathbb{E}_{p(\boldsymbol{\theta}|\boldsymbol{y})}\big[J_{\boldsymbol{\theta}}[\boldsymbol{u}]\big]$. Now by a similar calculation to that in the proof of Proposition 3.2, one can check that

$$\boldsymbol{u}_0^{(n+1)} = (1-\gamma)\boldsymbol{u}_0^{(n)} + \gamma\mathbf{B}^{-1}\mathbb{E}_{p(\boldsymbol{\theta}|\boldsymbol{y})}\Big[\mathbf{L}_{\boldsymbol{\theta}}^{(n)\top}\bar{\mathbf{Q}}_{\boldsymbol{\theta}}^{-1}\boldsymbol{m}_{\boldsymbol{\theta}}^{(n)} + \bar{\mathbf{H}}_{\boldsymbol{\theta}}^{\top}\bar{\mathbf{R}}_{\boldsymbol{\theta}}^{-1}\boldsymbol{y}\Big] \tag{78}$$

holds, where as before, we denoted

$$\boldsymbol{m}_{\boldsymbol{\theta}}^{(n)} := \mathbf{L}_{\boldsymbol{\theta}}^{(n)}\boldsymbol{u}_0^{(n)} - \mathcal{L}_{\boldsymbol{\theta}}[\boldsymbol{u}_0^{(n)}]. \tag{79}$$

Next, we claim that

$$\mathbf{B}^{-1}\mathbb{E}_{p(\boldsymbol{\theta}|\boldsymbol{y})}\Big[\mathbf{L}_{\boldsymbol{\theta}}^{(n)\top}\bar{\mathbf{Q}}_{\boldsymbol{\theta}}^{-1}\boldsymbol{m}_{\boldsymbol{\theta}}^{(n)} + \bar{\mathbf{H}}_{\boldsymbol{\theta}}^{\top}\bar{\mathbf{R}}_{\boldsymbol{\theta}}^{-1}\boldsymbol{y}\Big] \approx (\bar{\mathbf{P}}^{(n)})^{-1}\bar{\boldsymbol{b}}^{(n)}, \tag{80}$$

where

$$\bar{\mathbf{P}}^{(n)} := \sum_k \mathbf{P}_{\boldsymbol{u}|\boldsymbol{y}}^{(n)}(\boldsymbol{\theta}_k)\tilde{p}^{(n)}(\boldsymbol{\theta}_k|\boldsymbol{y})\Delta_k \tag{81}$$

$$\bar{\boldsymbol{b}}^{(n)} := \sum_k \mathbf{P}_{\boldsymbol{u}|\boldsymbol{y}}^{(n)}(\boldsymbol{\theta}_k)\boldsymbol{\mu}_{\boldsymbol{u}|\boldsymbol{y}}^{(n)}(\boldsymbol{\theta}_k)\tilde{p}^{(n)}(\boldsymbol{\theta}_k|\boldsymbol{y})\Delta_k, \tag{82}$$

To see this, recall from (70)–(71) that

$$\mathbf{P}_{\boldsymbol{u}|\boldsymbol{y}}^{(n)}(\boldsymbol{\theta}) = \bar{\mathbf{H}}_{\boldsymbol{\theta}}^{\top}\bar{\mathbf{R}}_{\boldsymbol{\theta}}^{-1}\bar{\mathbf{H}}_{\boldsymbol{\theta}} + \mathbf{L}_{\boldsymbol{\theta}}^{(n)\top}\bar{\mathbf{Q}}_{\boldsymbol{\theta}}^{-1}\mathbf{L}_{\boldsymbol{\theta}}^{(n)} \tag{83}$$

$$\boldsymbol{\mu}_{\boldsymbol{u}|\boldsymbol{y}}^{(n)}(\boldsymbol{\theta}) = \mathbf{P}_{\boldsymbol{u}|\boldsymbol{y}}^{(n)}(\boldsymbol{\theta})^{-1}\Big[\mathbf{L}_{\boldsymbol{\theta}}^{(n)\top}\bar{\mathbf{Q}}_{\boldsymbol{\theta}}^{-1}\boldsymbol{m}_{\boldsymbol{\theta}}^{(n)} + \bar{\mathbf{H}}_{\boldsymbol{\theta}}^{\top}\bar{\mathbf{R}}_{\boldsymbol{\theta}}^{-1}\boldsymbol{y}\Big]. \tag{84}$$

This gives us the approximations

$$\mathbf{B} \approx \mathbb{E}_{\tilde{p}^{(n)}(\boldsymbol{\theta}|\boldsymbol{y})}\big[\bar{\mathbf{H}}_{\boldsymbol{\theta}}^{\top}\bar{\mathbf{R}}_{\boldsymbol{\theta}}^{-1}\bar{\mathbf{H}}_{\boldsymbol{\theta}} + \mathbf{L}_{\boldsymbol{\theta}}^{(n)\top}\bar{\mathbf{Q}}_{\boldsymbol{\theta}}^{-1}\mathbf{L}_{\boldsymbol{\theta}}^{(n)}\big] \approx \bar{\mathbf{P}}^{(n)}, \quad \text{and} \tag{85}$$

$$\mathbb{E}_{p(\boldsymbol{\theta}|\boldsymbol{y})}\Big[\mathbf{L}_{\boldsymbol{\theta}}^{(n)\top}\bar{\mathbf{Q}}_{\boldsymbol{\theta}}^{-1}\boldsymbol{m}_{\boldsymbol{\theta}}^{(n)} + \bar{\mathbf{H}}_{\boldsymbol{\theta}}^{\top}\bar{\mathbf{R}}_{\boldsymbol{\theta}}^{-1}\boldsymbol{y}\Big] \approx \mathbb{E}_{\tilde{p}^{(n)}(\boldsymbol{\theta}|\boldsymbol{y})}\Big[\mathbf{L}_{\boldsymbol{\theta}}^{(n)\top}\bar{\mathbf{Q}}_{\boldsymbol{\theta}}^{-1}\boldsymbol{m}_{\boldsymbol{\theta}}^{(n)} + \bar{\mathbf{H}}_{\boldsymbol{\theta}}^{\top}\bar{\mathbf{R}}_{\boldsymbol{\theta}}^{-1}\boldsymbol{y}\Big] \approx \bar{\boldsymbol{b}}^{(n)}, \tag{86}$$

which proves our claim. Hence, we have shown that

$$(78) \approx (1-\gamma)\boldsymbol{u}_0^{(n)} + \gamma(\bar{\mathbf{P}}^{(n)})^{-1}\bar{\boldsymbol{b}}^{(n)}, \tag{87}$$

where the RHS is precisely the update rule (36)–(39). Note that for the approximations to be accurate, we require that (i) the estimates $\tilde{p}^{(n)}(\boldsymbol{\theta}|\boldsymbol{y})$ are close to the true posteriors $p(\boldsymbol{\theta}|\boldsymbol{y})$, and (ii) the numerical integrals (81) and (82) approximate closely the quantities $\mathbb{E}_{p(\boldsymbol{\theta}|\boldsymbol{y})}[\bar{\mathbf{P}}_{\boldsymbol{u}|\boldsymbol{y}}^{(n)}(\boldsymbol{\theta})]$ and $\mathbb{E}_{p(\boldsymbol{\theta}|\boldsymbol{y})}[\bar{\boldsymbol{b}}_{\boldsymbol{u}|\boldsymbol{y}}^{(n)}(\boldsymbol{\theta})]$, respectively. $\square$

## B.3 FURTHER DISCUSSION OF THE RESULTS

Here we provide discussion about interpretations and further error analysis of our results.

### B.3.1 Proposition 3.2

This result shows that at convergence, the linearisation point $\boldsymbol{u}_0^{(\infty)}$ is the MAP estimate of $p(\boldsymbol{u}|\boldsymbol{y}, \boldsymbol{\theta})$. This is also true for the corresponding posterior mean $\boldsymbol{\mu}_{\boldsymbol{u}|\boldsymbol{y}}^{(\infty)}(\boldsymbol{\theta})$, which we can show is identical to $\boldsymbol{u}_0^{(\infty)}$ (to see this, take $\boldsymbol{u}_0^{(n+1)} = \boldsymbol{u}_0^{(n)} = \boldsymbol{u}_0^{(\infty)}$ in the update formula (30)). Furthermore, the converged posterior precision reads (from (70))

$$\mathbf{P}_{\boldsymbol{u}|\boldsymbol{y}}^{(\infty)}(\boldsymbol{\theta}) = \bar{\mathbf{H}}^{\top}\bar{\mathbf{R}}^{-1}\bar{\mathbf{H}} + \mathbf{L}^{(\infty)\top}\bar{\mathbf{Q}}^{-1}\mathbf{L}^{(\infty)}, \tag{88}$$

where $\mathbf{L}^{(\infty)} := \nabla\mathcal{L}[\boldsymbol{u}_0^{(\infty)}]$. This is an approximation to the Hessian of the 4D-Var cost $J = -\log p(\boldsymbol{u}|\boldsymbol{y}, \boldsymbol{\theta})$:

$$\nabla^2 J[\boldsymbol{u}_0^{(\infty)}] = \bar{\mathbf{H}}^\top \bar{\mathbf{R}}^{-1} \bar{\mathbf{H}} + \mathbf{L}^{(\infty)\top} \bar{\mathbf{Q}}^{-1} \mathbf{L}^{(\infty)} + \nabla^2 \mathcal{L}[\boldsymbol{u}_0^{(\infty)}] \bar{\mathbf{Q}}^{-1} \mathcal{L}[\boldsymbol{u}_0^{(\infty)}], \tag{89}$$

which we refer to as the Gauss-Newton approximation of the Hessian. The only difference with the true Hessian is the term $\nabla^2 \mathcal{L}[\boldsymbol{u}_0^{(\infty)}] \bar{\mathbf{Q}}^{-1} \mathcal{L}[\boldsymbol{u}_0^{(\infty)}]$, which is small if $\nabla^2 \mathcal{L}[\boldsymbol{u}_0^{(\infty)}]$ or $\mathcal{L}[\boldsymbol{u}_0^{(\infty)}]$ is small. The former holds if the dynamics is weakly nonlinear and the latter holds if $\boldsymbol{u}_0^{(\infty)}$ is close to the solution of the deterministic system starting from $\boldsymbol{u}_0 = \boldsymbol{u}_b$ (see (55)). Now, the Laplace approximation for the distribution $p(\boldsymbol{u}|\boldsymbol{y}, \boldsymbol{\theta})$ is given by

$$p(\boldsymbol{u}|\boldsymbol{y}, \boldsymbol{\theta}) \approx \mathcal{N}(\boldsymbol{u}|\boldsymbol{u}_*, \nabla^2 J[\boldsymbol{u}_*]^{-1}), \quad \text{where} \quad \boldsymbol{u}_* := \operatorname{argmin}_{\boldsymbol{u}}\{J[\boldsymbol{u}]\}. \tag{90}$$

Hence, assuming that $\mathbf{P}_{\boldsymbol{u}|\boldsymbol{y}}^{(\infty)}(\boldsymbol{\theta}) \approx \nabla^2 J[\boldsymbol{u}_*]$, we see that the outputs of Algorithm 1 can be interpreted as the marginals of a "Gauss-Newton-Laplace" approximation of the posterior $p(\boldsymbol{u}|\boldsymbol{y}, \boldsymbol{\theta})$.

### B.3.2 Proposition 3.3

In this result, we see that the converged point $\boldsymbol{u}_0^{(\infty)}$ using type-II iterated INLA is an approximate MAP estimate of the marginal posterior $p(\boldsymbol{u}|\boldsymbol{y})$, whose log-transform is lower bounded by a surrogate cost $\mathbb{E}_{p(\boldsymbol{\theta}|\boldsymbol{y})}[\log p(\boldsymbol{u}|\boldsymbol{y}, \boldsymbol{\theta})]$ that is being optimised by the algorithm. One can check that the gap between the two quantities can be characterised exactly as

$$\log p(\boldsymbol{u}|\boldsymbol{y}) - \mathbb{E}_{p(\boldsymbol{\theta}|\boldsymbol{y})}[\log p(\boldsymbol{u}|\boldsymbol{y}, \boldsymbol{\theta})] = \mathcal{KL}(p(\boldsymbol{\theta}|\boldsymbol{y})||p(\boldsymbol{\theta}|\boldsymbol{u}, \boldsymbol{y})). \tag{91}$$

Thus, the mode $\boldsymbol{u}_*$ of $\mathbb{E}_{p(\boldsymbol{\theta}|\boldsymbol{y})}[\log p(\boldsymbol{u}|\boldsymbol{y}, \boldsymbol{\theta})]$ is close to the mode of $\log p(\boldsymbol{u}|\boldsymbol{y})$ provided $\mathcal{KL}(p(\boldsymbol{\theta}|\boldsymbol{y})||p(\boldsymbol{\theta}|\boldsymbol{u}_*, \boldsymbol{y})) \approx 0$. To see this, assuming $\mathcal{KL}(p(\boldsymbol{\theta}|\boldsymbol{y})||p(\boldsymbol{\theta}|\boldsymbol{u}_*, \boldsymbol{y})) = 0$, we have $\nabla_{\boldsymbol{u}} \mathcal{KL}(p(\boldsymbol{\theta}|\boldsymbol{y})||p(\boldsymbol{\theta}|\boldsymbol{u}, \boldsymbol{y}))|_{\boldsymbol{u}=\boldsymbol{u}_*} = \boldsymbol{0}$ since the KL-divergence is always non-negative. Hence the gradient of $\log p(\boldsymbol{u}|\boldsymbol{y})$ also vanishes at $\boldsymbol{u}_*$, making it a mode.

The assumption that $\mathcal{KL}(p(\boldsymbol{\theta}|\boldsymbol{y})||p(\boldsymbol{\theta}|\boldsymbol{u}_*, \boldsymbol{y})) \approx 0$ should hold if for example $p(\boldsymbol{\theta}|\boldsymbol{y})$ is very peaked and depends weakly on $\boldsymbol{u}$. In this case, the MAP estimate of $p(\boldsymbol{u}|\boldsymbol{y})$ should be reliably approximated by $\boldsymbol{u}_0^{(\infty)}$.

## C EXPERIMENT DETAILS

### C.1 METRICS

In our experiments, we use the following metrics to benchmark our results.

**Root Mean Square Error (RMSE):** The root mean squared error quantifies the average deviation of an estimate of a quantity from its ground truth value. Denoting by $\boldsymbol{u}^{gt} \in \mathbb{R}^{d_u}$ the ground truth and $\hat{\boldsymbol{u}} \in \mathbb{R}^{d_u}$ our estimate for it, then the RMSE is computed as follows.

$$\text{RMSE}(\boldsymbol{u}^{gt}, \hat{\boldsymbol{u}}) = \sqrt{\frac{1}{d_u} \sum_{i=1}^{d_u} \|\hat{u}_i - u_i^{gt}\|^2} \tag{92}$$

The choice of the estimated quantity $\hat{\boldsymbol{u}}$ depends on our model. For instance, if the outputs are Gaussian, then a sensible choice is its mean, or if its non-Gaussian, then we may also choose its median or mode. For iterated INLA, we choose the converged linearisation points $\boldsymbol{u}_0^{(\infty)}$ as our estimator since by Propositions 3.2 and 3.3, these approximate the mode of the corresponding distributions.

**Mean Negative Log-Likelihood (MNLL):** Another useful metric to use is the negative log-likelihood, which also evaluates the quality of uncertainties produced by our models. This is computed as

$$\text{MNLL}(\boldsymbol{u}^{gt}, \tilde{p}(u_i|\boldsymbol{y})) = \frac{1}{d_u} \sum_{i=1}^{d_u} \left( -\log \tilde{p}(u_i|\boldsymbol{y}) \right)\Big|_{u_i = u_i^{gt}}, \tag{93}$$

where $\tilde{p}(u_i|\boldsymbol{y})$ are the estimated marginal posteriors from our inference methods.

**Maximum Mean Discrepancy (MMD):** The maximum mean discrepancy (MMD) compares the similarity of two probability distributions $\pi_1$ and $\pi_2$ based on their samples. Given samples $\boldsymbol{u}_n \sim \pi_1$ for $n = 1, \ldots, N$ and $\boldsymbol{v}_m \sim \pi_2$ for $m = 1, \ldots, M$, the MMD between $\pi_1$ and $\pi_2$ is computed as [Gretton et al., 2012]

$$\text{MMD}(\{\boldsymbol{u_n}\}_{n=1}^N, \{\boldsymbol{v}_m\}_{m=1}^M) = \frac{1}{N}\frac{1}{(N-1)} \sum_{n=1}^N \sum_{m=1}^N k(\boldsymbol{u}_n, \boldsymbol{u}_m) - 2\frac{1}{NM} \sum_{n=1}^N \sum_{m=1}^M k(\boldsymbol{u}_n, \boldsymbol{v}_m)$$
$$+ \frac{1}{M}\frac{1}{(M-1)} \sum_{n=1}^M \sum_{m=1}^M k(\boldsymbol{v}_n, \boldsymbol{v}_m). \tag{94}$$

Here, $k(\cdot, \cdot)$ is a kernel, which we choose to be squared exponential by default. In our pendulum experiment, each sample $\boldsymbol{u}_n$ is a vector whose $i$-th component is a sample from the marginal posterior $p(u_i|\boldsymbol{y})$. Computing the MMD for every time slice and taking its average can be very time consuming, so instead we compute the MMD once on the product distribution $\prod_{i=1}^{N_t} p(u_i|\boldsymbol{y})$ with a kernel defined on $\mathbb{R}^{N_t}$. For this, one must be careful to avoid having correlations between two consecutive time steps. For instance, a sample trajectory from a particle smoother or the ensemble Kalman smoother will have strong correlations between consecutive timesteps since these are samples from the *joint* distribution $p(\boldsymbol{u}|\boldsymbol{y})$. Thus in these situations, one must ensure to scramble the particles at each time step before computing the MMD to ensure correct sampling from the product distribution $\prod_{i=1}^{N_t} p(u_i|\boldsymbol{y})$.

## C.2 STOCHASTIC PENDULUM EXPERIMENT

Here, we provide details on the pendulum experiment presented in Section 4.1.

### C.2.1 Model configuration

The stochastic nonlinear pendulum system is described by the equation

$$\frac{\mathrm{d}^2 u}{\mathrm{d}t^2} + b\frac{\mathrm{d}u}{\mathrm{d}t} + c\sin(u) = \sigma_u \dot{W}_t. \tag{95}$$

Here, $b, c > 0$ are some constants describing the damping and forcing rates resepectively, $\sigma_u > 0$ is the process noise amplitude and $W_t$ is a 1D Wiener process. This describes the nonlinear dynamics of a damped pendulum, oscillating under the influence of gravity and continuously perturbed by random forces.

More rigorously, we interpret the equation (95) as a coupled first-order Itô diffusion process

$$\begin{cases} \mathrm{d}u &= \omega\,\mathrm{d}t \\ \mathrm{d}\omega &= -b\omega\,\mathrm{d}t - c\sin(u)\,\mathrm{d}t + \sigma_u\,\mathrm{d}W_t. \end{cases} \tag{96}$$

Here, $u \in [-\pi, \pi]$ describes the dynamics of the angle of the pendulum and $\omega \in \mathbb{R}$ is the angular velocity of the system. For the ground truth, we simulated the dynamics of (96) on various random seeds starting from $u(0) = 0.75\pi, u'(0) = 0$ and ran for $t \in [0, 25]$ using the Euler-Maruyama scheme with a timestep of $\Delta t = 0.01$. We fixed the values $b = 0.3$, $c = 1.0$ and $\sigma_u = 0.2$ throughout the experiment. For the observations $\boldsymbol{y}$, we randomly selected 5% of gridpoints $t_n$ within the time interval $[0, 10]$, then sampled from i.i.d. Gaussians $y_n \sim \mathcal{N}(u(t_n), \sigma_y^2)$ with observation noise $\sigma_y = 0.1$.

For the priors on the parameters, we used log normal distributions in order to ensure positivity. In particular, we took

$$b \sim \text{LogNormal}(-1.36, 0.5), \tag{97}$$
$$c \sim \text{LogNormal}(1.69, 1.0), \tag{98}$$
$$\sigma_u \sim \text{LogNormal}(-2.05, 0.5), \tag{99}$$
$$\sigma_y \sim \text{LogNormal}(-2.05, 0.5). \tag{100}$$

Note that $z \sim \text{LogNormal}(\mu, \sigma)$ means $z = e^x$ for $x \sim \mathcal{N}(\mu, \sigma^2)$. The modes of the distributions are $0.2, 2.0, 0.1$ and $0.1$ respectively.

### C.2.2 Linearisation

To linearise the system (95) around a point $u_0$, let $|u - u_0| = \mathcal{O}(\epsilon)$ for $\epsilon \ll 1$. Then by Taylor expansion, we have

$$\sin(u) = \sin(u_0) + \cos(u_0)(u - u_0) + \mathcal{O}(\epsilon^2). \tag{101}$$

Substituting this into the LHS of (95), we get

$$\mathcal{L}[u] := \frac{\mathrm{d}^2 u}{\mathrm{d}t^2} + b\frac{\mathrm{d}u}{\mathrm{d}t} + c\sin(u) \tag{102}$$

$$\approx \frac{\mathrm{d}^2 u}{\mathrm{d}t^2} + b\frac{\mathrm{d}u}{\mathrm{d}t} + c\big(\sin(u_0) + \cos(u_0)(u - u_0)\big) + \mathcal{O}(\epsilon^2) \tag{103}$$

$$= \left(\frac{\mathrm{d}^2 u}{\mathrm{d}t^2} + b\frac{\mathrm{d}u}{\mathrm{d}t} + c\cos(u_0)u\right) - c\big(u_0\cos(u_0) - \sin(u_0)\big) + \mathcal{O}(\epsilon^2). \tag{104}$$

This gives us

$$\mathcal{L}_0 u = \frac{\mathrm{d}^2 u}{\mathrm{d}t^2} + b\frac{\mathrm{d}u}{\mathrm{d}t} + c\cos(u_0)u \quad \text{and} \quad r_0 = c\big(u_0\cos(u_0) - \sin(u_0)\big). \tag{105}$$

### C.2.3 Settings for iterated INLA

We used the zero function $u(t) \equiv 0$ as the initial linearisation point $u_0^{(0)}$ and set the damping rate to $\gamma = 0.3$. For the acceptance threshold in (51), we chose $\delta = 5$. We set the number of iterations to 25. To discretise the linear model (105), we used centered finite differences with a grid size of $\Delta t = 0.01$. See Appendix D for more details on the discretisation.

### C.2.4 Baseline details

Here, we provide further details on the baseline models we used for comparison.

**Sequential Monte-Carlo (SMC).** For our SMC method, we split the procedure into two parts. The first part samples the parameters $\boldsymbol{\theta}_k \sim p(\boldsymbol{\theta}|\boldsymbol{y})$ using the particle marginal Metropolis-Hastings (PMMH) method in Andrieu et al. [2010]. This is a Metropolis-Hastings algorithm that approximates the term $p(\boldsymbol{y}|\boldsymbol{\theta})$ in the acceptance ratio $p(\boldsymbol{y}|\boldsymbol{\theta}')p(\boldsymbol{\theta}')/p(\boldsymbol{y}|\boldsymbol{\theta}_n)p(\boldsymbol{\theta}_n)$ using a bootstrap particle filter with fixed $\boldsymbol{\theta}$. We used $1,000$ particles for the bootstrap filter and sampled $10,000$ parameters from $p(\boldsymbol{\theta}|\boldsymbol{y})$. We used a burn-in period of $1,000$. In the next step, we used the generated parameters $\boldsymbol{\theta}_k$ and computed a sample $\boldsymbol{u}_k$ of $p(\boldsymbol{u}|\boldsymbol{\theta}_k, \boldsymbol{y})$ for each $k$ using the bootstrap particle smoother [Chopin and Papaspiliopoulos, 2020], which are precisely the samples of $p(\boldsymbol{u}|\boldsymbol{y})$. The Euler-Maruyama scheme was used to simulate the dynamics (96) with a time step of $\Delta t = 0.01$. For the implementation of PMMH and the bootstrap particle filter/smoother, we used the python package `particles`[1].

**Gaussian process regression (GPR).** We used a Gaussian process with the standard RBF kernel (i.e., squared exponential kernel) initialised with unit lenghscale and amplitude. The standard deviation in the Gaussian likelihood was initialised at 0.1. The hyperparameters were tuned via type-II maximum likelihood estimation, performed using the L-BFGS-B optimiser.

**Ensemble Kalman Smoother (EnKS).** We used the EnKS implementation in DAPPER Raanes et al. [2018], which by default, uses the deterministic (i.e., Ensemble Transform) variant of the EnKS, typically considered state of the art. We used 100 ensemble members with no inflation; we found that the inflation caused instability when learning the noise parameter $\sigma_u$ and found better results without it. We used the Euler-Maruyama discretisation with a time step of $\Delta t = 0.01$ to propagate the dynamics of (96) forward in time. We also evolved the parameters by persistent dynamics, i.e., $\boldsymbol{\theta}_{n+1} = \boldsymbol{\theta}_n$ to jointly infer the state and parameters using the state-augmentation method Evensen [2009]. The parameters were propagated in log-space to retain positivity. This also made it consistent with the log-normal priors used for the parameters; at initialisation, the parameters were sampled exactly from (97)–(100). For the initial state, we used a Gaussian around the true initial values with a standard deviation of 0.1.

---

[1] https://github.com/nchopin/particles

**Iterated Ensemble Kalman Smoother (iEnKS).** For a single iteration of the EnKS, we used the same configuration of EnKS as above. The result is displayed for 10 iterations.

**AutoIP.** For this experiment, we used the original pendulum code accompanying Long et al. [2022] to generate our results[2]. In particular, they conduct a similar pendulum experiment in their work and we used the same code without modification. To initialise the hyperparameters $b, c, \sigma_u$ and $\sigma_y$, we used the modal values of the respective priors. All of the parameters were learned alongside the variational parameters in the variational inference employed in AutoIP. We trained the model for $1500$ epochs with early stopping and the optimisation was performed using Adam with the default parameters and a learning rate set to $0.01$.

## C.3 BURGERS' EXPERIMENT

The 1D Burgers' equation is given by

$$\frac{\partial u}{\partial t} + u\frac{\partial u}{\partial x} - \nu\frac{\partial^2 u}{\partial x^2} = 0, \tag{106}$$

where $\nu > 0$ is the viscosity parameter. We assume periodic boundary conditions on $x \in [-1, 1]$ and used the initial condition $u(x, 0) = -\sin(\pi x)$. To generate the ground truth, we used the pseudospectral method of lines, integrating between the period $t \in [0, 0.5]$ with a time step of $\Delta t = 0.01$ and setting the viscosity parameter to $\nu = 0.02$.

For the observations, we uniformly sampled 20 observations each from the ground truth along two strips, one at $t = 0$ and another at $t = 0.26$, for a total of $40$ observation points. These were then perturbed independently by centered Gaussian observation noise with a standard deviation of $0.1$.

### C.3.1 Linearisation

Take $u = u_0 + \epsilon v$ for $\epsilon \ll 1$. Then, the nonlinear advection term in (106) can be linearised as follows

$$u\frac{\partial u}{\partial x} = (u_0 + \epsilon\, v)\frac{\partial}{\partial x}(u_0 + \epsilon\, v) \tag{107}$$

$$= u_0\frac{\partial u_0}{\partial x} + \epsilon\left(u_0\frac{\partial v}{\partial x} + v\frac{\partial u_0}{\partial x}\right) + \mathcal{O}(\epsilon^2) \tag{108}$$

$$= u_0\frac{\partial u_0}{\partial x} + u_0\frac{\partial}{\partial x}(\epsilon v) + (\epsilon v)\frac{\partial u_0}{\partial x} + \mathcal{O}(\epsilon^2) \tag{109}$$

$$= u_0\frac{\partial u_0}{\partial x} + u_0\frac{\partial}{\partial x}(u - u_0) + (u - u_0)\frac{\partial u_0}{\partial x} + \mathcal{O}(\epsilon^2) \tag{110}$$

$$= u_0\frac{\partial u_0}{\partial x} - u_0\frac{\partial u_0}{\partial x} + u_0\frac{\partial u}{\partial x} + u\frac{\partial u_0}{\partial x} - u_0\frac{\partial u_0}{\partial x} + \mathcal{O}(\epsilon^2) \tag{111}$$

$$= u_0\frac{\partial u}{\partial x} + u\frac{\partial u_0}{\partial x} - u_0\frac{\partial u_0}{\partial x} + \mathcal{O}(\epsilon^2). \tag{112}$$

Plugging this back into the LHS of (106), we get

$$\mathcal{L}[u] = \frac{\partial u}{\partial t} + u\frac{\partial u}{\partial x} - \nu\frac{\partial^2 u}{\partial x^2} \tag{113}$$

$$= \frac{\partial u}{\partial t} + \left(u_0\frac{\partial u}{\partial x} + u\frac{\partial u_0}{\partial x} - u_0\frac{\partial u_0}{\partial x}\right) - \nu\frac{\partial^2 u}{\partial x^2} + \mathcal{O}(\epsilon^2) \tag{114}$$

$$= \left(\frac{\partial u}{\partial t} + u_0\frac{\partial u}{\partial x} + u\frac{\partial u_0}{\partial x} - \nu\frac{\partial^2 u}{\partial x^2}\right) - u_0\frac{\partial u_0}{\partial x} + \mathcal{O}(\epsilon^2). \tag{115}$$

Thus, we have

$$\mathcal{L}_0 u = \frac{\partial u}{\partial t} + u_0\frac{\partial u}{\partial x} + u\frac{\partial u_0}{\partial x} - \nu\frac{\partial^2 u}{\partial x^2} \quad \text{and} \quad r_0 = u_0\frac{\partial u_0}{\partial x}. \tag{116}$$

---

[2]`https://github.com/long-da/A-United-Framework-to-Integrate-Physics-into-Gaussian-Processes`

### C.3.2 Settings for iterated INLA

We imposed priors on the viscosity parameter $\nu$ and the process noise amplitude $\sigma_u$. Again, we used log normal distributions in order to ensure positivity of the parameters. In particular, we took

$$\nu \sim \text{LogNormal}(-2.0, 1.0), \tag{117}$$

$$\sigma_u \sim \text{LogNormal}(-3.6, 1.0), \tag{118}$$

$$\tag{119}$$

which has modes $0.05$ and $0.01$ respectively. The linearisation point was initialised by a solution to the Burgers' equation with initial condition $u_b$ (the background field) and the parameter $\nu$ set to $0.05$, i.e., the mode of the prior on $\nu$, not the ground truth value of $0.01$. The background field $u_b$ was taken as the prediction from the GPR baseline (see Appendix C.3.3) at time $t = 0$. We set the number of iterations to 10. For this experiment, we used a damping rate of $\gamma = 0.5$ and acceptance threshold of $\delta = 3$. Discretisation was performed using the central finite difference scheme with $\Delta t = 0.02$ and $\Delta x = 0.04$.

### C.3.3 Baseline details

**GPR.** We used the same GPR setting as described in Appendix C.2.4.

**EnKS.** We used 100 ensemble members with no inflation. For the time-stepping, we used a variant of the fourth-order Runge-Kutta scheme in Kassam and Trefethen [2005] with a time step of $\Delta t = 10^{-3}$ and a spatial step of $\Delta x = 0.04$. We found that the standard Runge-Kutta scheme led to unstable solutions, especially when jointly learning the parameter.

**iEnKS.** For a single iteration of the EnKS, we used the same configuration of EnKS as above. We used 30 iterations to produce the final result.

**AutoIP.** We adapted the original AutoIP code to accommodate the Burgers' system. We trained the $\nu$ parameter and the parameter corresponding to our noise process $\sigma_u$, setting the initial values to the mode of the respective priors (117)–(118). We set the initial lengthscales of the latent GP to $l_x = 0.5$ and $l_t = 0.5$. The model was trained for 2000 epochs with early stopping and optimisation was performed with Adam with the learning rate set to $0.01$.

## C.4 ALLEN-CAHN EXPERIMENT

The 1D Allen-Cahn equation is given by

$$\frac{\partial u}{\partial t} - \gamma \frac{\partial^2 u}{\partial x^2} + f(u) = 0, \tag{120}$$

where $f(u)$ is the source term, which we take to be $f(u) = \beta(u^3 - u)$ for some $\beta > 0$. For the ground truth, we used the pre-computed simulation found in the PINNs GitHub repository[3]. This has the configuration $\beta = 5$, $\gamma = 10^{-4}$, with periodic boundary conditions and an initial condition set to $u(x, 0) = x^2 \cos(\pi x)$. The simulation is for times $t \in [0, 1]$ with a spatial domain of size $[-1, 1]$.

We sampled 256 random observation from the ground truth from a uniform distribution in $(t, x) \in [0, 0.28] \times [-1, 1]$. The values were then perturbed independently by a centered Gaussian noise with standard deviation $\sigma_y = 0.01$.

### C.4.1 Linearisation

We approximate the nonlinear term $u^3$ around a point $u_0$ by Taylor expansion

$$u^3 \approx u_0^3 + 3u_0^2(u - u_0) + \mathcal{O}(\epsilon^2). \tag{121}$$

---

[3]https://github.com/maziarraissi/PINNs/tree/master

Substituting this expression into the LHS of (120), we get

$$\mathcal{L}[u] = \frac{\partial u}{\partial t} - \gamma \frac{\partial^2 u}{\partial x^2} + \beta(u_0^3 + 3u_0^2(u - u_0)) - \beta u + \mathcal{O}(\epsilon^2) \tag{122}$$

$$= \left( \frac{\partial u}{\partial t} - \gamma \frac{\partial^2 u}{\partial x^2} + 3\beta u_0^2 u - \beta u \right) - 2\beta u_0^3 + \mathcal{O}(\epsilon^2). \tag{123}$$

Hence, we have

$$\mathcal{L}_0 u = \frac{\partial u}{\partial t} - \gamma \frac{\partial^2 u}{\partial x^2} + 3\beta u_0^2 u - \beta u \quad \text{and} \quad r_0 = 2\beta u_0^3. \tag{124}$$

### C.4.2 Settings for iterated INLA.

We imposed priors on the $\beta$ parameter and the process noise parameter $\sigma_u$. The $\gamma$ parameter and the observation noise parameter $\sigma_y$ was held fixed. For the trainable parameters, we took the priors

$$\beta \sim \text{LogNormal}(2.10, 1.0), \tag{125}$$
$$\sigma_u \sim \text{LogNormal}(-3.60, 1.0), \tag{126}$$
$$\tag{127}$$

which has modes 3.0 and 0.01, respectively. As with the Burgers' experiment, we initialised the linearisation point by a solution to the Allen-Cahn system with initial condition $u_b$, obtained by the prediction of the GPR baseline at $t = 0$, and the learnable parameters set to its respective prior mode. We set the number of iterations to 10. For discretisation of the linearised operator, we used a vanilla central finite difference scheme with $\Delta t = 0.02$ and $\Delta x = 1/64$.

### C.4.3 Baseline details

**GPR.** We used the same GPR setting as described in Appendix C.2.4.

**EnKS.** We used 100 ensemble members with no inflation. The time stepping was performed using the same RK4 solver that we used for the EnKS baseline in the Burger's experiment (Appendix C.3.3) with a time step of $\Delta t = 0.005$ and a spatial step of $\Delta x = 1/64$.

**iEnKS.** For a single iteration of the EnKS, we used the same configuration of EnKS as above. We used 30 iterations to produce the final result.

**AutoIP.** The AutoIP code contained an Allen-Cahn example, which we used unchanged. We trained the $\beta$ parameter and the parameter corresponding to our noise process $\sigma_u$, setting the initial values to the mode of the respective priors (125)–(126). We set the initial lengthscales of the latent GP to $l_x = 1.0$ and $l_t = 1.0$. The model was trained for 2000 epochs with early stopping and optimisation was performed with Adam with the learning rate set to 0.01.

### C.5 KORTEWEG-DE VRIES EXPERIMENT

The Korteweg-de Vries (KdV) equation is given by

$$\frac{\partial u}{\partial t} + \lambda_1 u \frac{\partial u}{\partial x} + \lambda_2 \frac{\partial^3 u}{\partial x^3} = 0, \tag{128}$$

modelling shallow water waves. Here, $\lambda_1$ and $\lambda_2$ are positive constants modelling the advection strength and dispersion rates respectively. Again, we used the pre-computed simulation found in the PINNs GitHub repository, which uses the configuration $\lambda_1 = 1.0, \lambda_2 = 0.0025$, periodic boundary condition and an initial condition of $u(x, 0) = \cos(\pi x)$. The simulation spans a time interval of $t \in [0, 1]$ and the spatial domain has size $x \in [-1, 1]$.

For the observations, we uniformly sampled 20 observations each from the ground truth along two strips, one at $t = 0.2$ and another at $t = 0.8$, for a total of 40 observation points. These were then perturbed independently by centered Gaussian observation noise with a standard deviation of $10^{-3}$.

### C.5.1 Linearisation

The linearisation procedure for the KdV equation is identical to that for the Burgers' equation so we refer the readers to section C.3.1 for the details. The resulting linearisation of the KdV equation (128) reads

$$\mathcal{L}_0 u = \frac{\partial u}{\partial t} + \lambda_1 \left( u_0 \frac{\partial u}{\partial x} + u \frac{\partial u_0}{\partial x} \right) + \lambda_2 \frac{\partial^3 u}{\partial x^3} \quad \text{and} \quad r_0 = \lambda_1 u_0 \frac{\partial u_0}{\partial x}. \tag{129}$$

### C.5.2 Settings for iterated INLA.

We imposed priors on the $\lambda_1$ parameter and the process noise parameter $\sigma_u$. The $\lambda_2$ parameter and the observation noise parameter $\sigma_y$ was held fixed. For the trainable parameters, we took the priors

$$\lambda_1 \sim \text{LogNormal}(0.31, 1.0), \tag{130}$$
$$\sigma_u \sim \text{LogNormal}(-3.60, 1.0), \tag{131}$$
$$\tag{132}$$

which has modes $0.5$ and $0.01$, respectively. As with the previous experiments, we initialised the linearisation point by a solution to the KdV system with initial condition $u_b$, obtained by the prediction of the GPR baseline at $t = 0$, and the learnable parameters set to its respective prior mode. We set the number of iterations to 10. For the discretisation of the linearised operator, we used a vanilla central finite difference scheme with $\Delta t = 0.02$ and $\Delta x = 1/64$.

### C.5.3 Baseline details

**GPR.** We used the same GPR setting as described in Appendix C.2.4.

**EnKS.** We used 100 ensemble members with no inflation. The time stepping was performed using the same RK4 solver that we used in the previous experiments with a time step of $\Delta t = 0.005$ and a spatial step of $\Delta x = 1/64$.

**iEnKS.** For a single iteration of the EnKS, we used the same configuration of EnKS as above. We used 30 iterations to produce the final result.

**AutoIP.** We adapted the original AutoIP code to accommodate the KdV system. We trained the $\lambda_1$ parameter and the parameter corresponding to our noise process $\sigma_u$, setting the initial values to the mode of the respective priors (130)–(131). We set the initial lengthscales of the latent GP to $l_x = 0.01$ and $l_t = 0.1$. The model was trained for 2000 epochs with early stopping and optimisation was performed with Adam with the learning rate set to $0.01$.

## D DISCRETISATION DETAILS

All discretisations are performed using finite differences with the python package `findiff` [Baer, 2018]. We discretised the linearised operators (105), (116), (124) and (129) using second-order central finite differences, treating the spatial and temporal variables on the same footing (i.e., we do not consider forward time-stepping methods). We imposed appropriate boundary conditions depending on the problem. `findiff` implements Dirichlet and Neumann boundary conditions. However, some of the experiments require periodic boundary conditions, hence we added this functionality in our fork of the package, which we do not disclose here to preserve anonymity. On the temporal boundaries, `findiff` by default uses a forward discretisation of the derivative at the initial time and backward discretisation at the final time, if the conditions are not specified. While this does not make sense physically, we found that using this default set up gave us sufficiently good results for our purpose. The random initial conditions were specified by additional likelihoods at the initial time, where we placed pseudo-observations $p(\boldsymbol{y}|\boldsymbol{u}_0)$ at time $t = 0$ to mimic the initial condition prior $p(\boldsymbol{u}_0) = \mathcal{N}(\boldsymbol{u}_0|\boldsymbol{u}_b, \mathbf{C})$. Mathematically, this is not a problem by taking $p(\boldsymbol{y}|\boldsymbol{u}_0)|_{\boldsymbol{y}=\boldsymbol{u}_b} = \mathcal{N}(\boldsymbol{y}|\boldsymbol{u}_b, \mathbf{C})|_{\boldsymbol{y}=\boldsymbol{u}_b}$ as it results in the same posterior distribution. In the future, it might be interesting to encode the initial condition prior directly into the model, using proper temporal time-stepping schemes such as the Crank-Nicolson method to discretise parabolic PDEs.

For the spatio-temporal white noise process over a domain $[0, T] \times \mathbb{R}$, we use the discretisation

$$\dot{\mathcal{W}}^N(x, t) = \sum_{i=1}^{N} \frac{z_i}{\sqrt{\Delta x \Delta t}} \mathbf{1}_{C_i}(x, t), \quad z \sim \mathcal{N}(0, \mathbf{I}), \tag{133}$$

where $C_i \subset [0, T] \times \mathbb{R}$ is an individual cell of a finite difference discretised spatio-temporal domain and $\mathbf{1}_{C_i}$ is the indicator function. To justify this, we use the following definition of the Gaussian white noise process.

**Definition D.1** (Lototsky and Rozovsky [2017], Definition 3.2.10). A (centered) generalised Gaussian white noise process $\mathcal{B}$ over a Hilbert space $H$ is a collection of random variables $\mathcal{B} \in H^*$, such that

1. For every $f \in H$, we have $\mathcal{B}f = 0$.
2. For every $f, g \in H$, we have $\mathbb{E}[\mathcal{B}f\,\mathcal{B}g] = \langle f, g \rangle_H$.

In particular, taking $H = L^2([0, T] \times \mathbb{R}; \mathbb{R})$, we arrive at the space-time white noise process $\mathcal{B} = \dot{\mathcal{W}}$ that we consider here. To see informally that $\dot{\mathcal{W}}^N$ approximates $\dot{\mathcal{W}}$, for every $f \in L^2([0, T] \times \mathbb{R}, \mathbb{R})$, we have

$$\left\langle \dot{\mathcal{W}}^N, f \right\rangle_{L^2} = \sum_{i=1}^{N} \frac{z_i}{\sqrt{\Delta x \Delta t}} \iint_{C_i} f(x, t) \mathrm{d}x \mathrm{d}t. \tag{134}$$

Furthermore, for small $|C_i| = \Delta x \Delta t$, we have

$$\iint_{C_i} f(x, t) \mathrm{d}x \mathrm{d}t \approx f(x_i, t_i) \Delta x \Delta t, \tag{135}$$

for any $(x_i, t_i) \in C_i$. Thus, we have the approximation

$$\left\langle \dot{\mathcal{W}}^N, f \right\rangle_{L^2} \approx \sum_{i=1}^{N} \frac{z_i f(x_i, t_i)}{\sqrt{\Delta x \Delta t}} \Delta x \Delta t = \sum_{i=1}^{N} z_i f(x_i, t_i) \sqrt{\Delta x \Delta t} \tag{136}$$

and we see that $\left\langle \mathcal{W}^N, h \right\rangle_{L^2}$ is Gaussian with moments

$$\mathbb{E}\left[ \left\langle \dot{\mathcal{W}}^N, f \right\rangle_{L^2} \right] = \sum_{i=1}^{N} \underbrace{\mathbb{E}[z_i]}_{=0} f(x_i, t_i) \sqrt{\Delta x \Delta t} = 0 \tag{137}$$

$$\mathbb{E}\left[ \left\langle \dot{\mathcal{W}}^N, f \right\rangle_{L^2} \left\langle \dot{\mathcal{W}}^N, g \right\rangle_{L^2} \right] = \sum_{i=1}^{N} \sum_{j=1}^{N} \underbrace{\mathbb{E}[z_i z_j]}_{=\delta_{ij}} f(x_i, t_i) g(x_j, t_j) \Delta x \Delta t = \sum_{i=1}^{N} f(x_i, t_i) g(x_i, t_i) \Delta x \Delta t. \tag{138}$$

Taking $N \to \infty$, these converge as

$$\mathbb{E}\left[ \left\langle \dot{\mathcal{W}}^N, f \right\rangle_{L^2} \right] \to 0 \tag{139}$$

$$\mathbb{E}\left[ \left\langle \dot{\mathcal{W}}^N, f \right\rangle_{L^2} \left\langle \dot{\mathcal{W}}^N, g \right\rangle_{L^2} \right] = \sum_{i=1}^{N} f(x_i, t_i) g(x_i, t_i) \Delta x \Delta t \to \iint f(x, t) g(x, t) \mathrm{d}x \mathrm{d}t = \langle f, g \rangle_{L^2}, \tag{140}$$

where the latter follows from the definition of Riemann integration. Thus, the moments of $\left\langle \dot{\mathcal{W}}^N, f \right\rangle_{L^2}$ converge to the moments of $\dot{\mathcal{W}}f$ as $N \to \infty$ and since $f, g \in L^2([0, T] \times \mathbb{R}, \mathbb{R})$ were chosen arbitrarily, we have the convergence in law

$$\dot{\mathcal{W}}^N \to \dot{\mathcal{W}}, \tag{141}$$

which is sufficient for our purpose.

# E  VISUALISATION OF RESULTS

In this appendix, we plot the results produced by iterated INLA and all the baseline models considered in the PDE benchmark experiments (Section 4.2). We display the predicted means and standard deviations for each method.

## E.1  BURGERS' EXPERIMENT

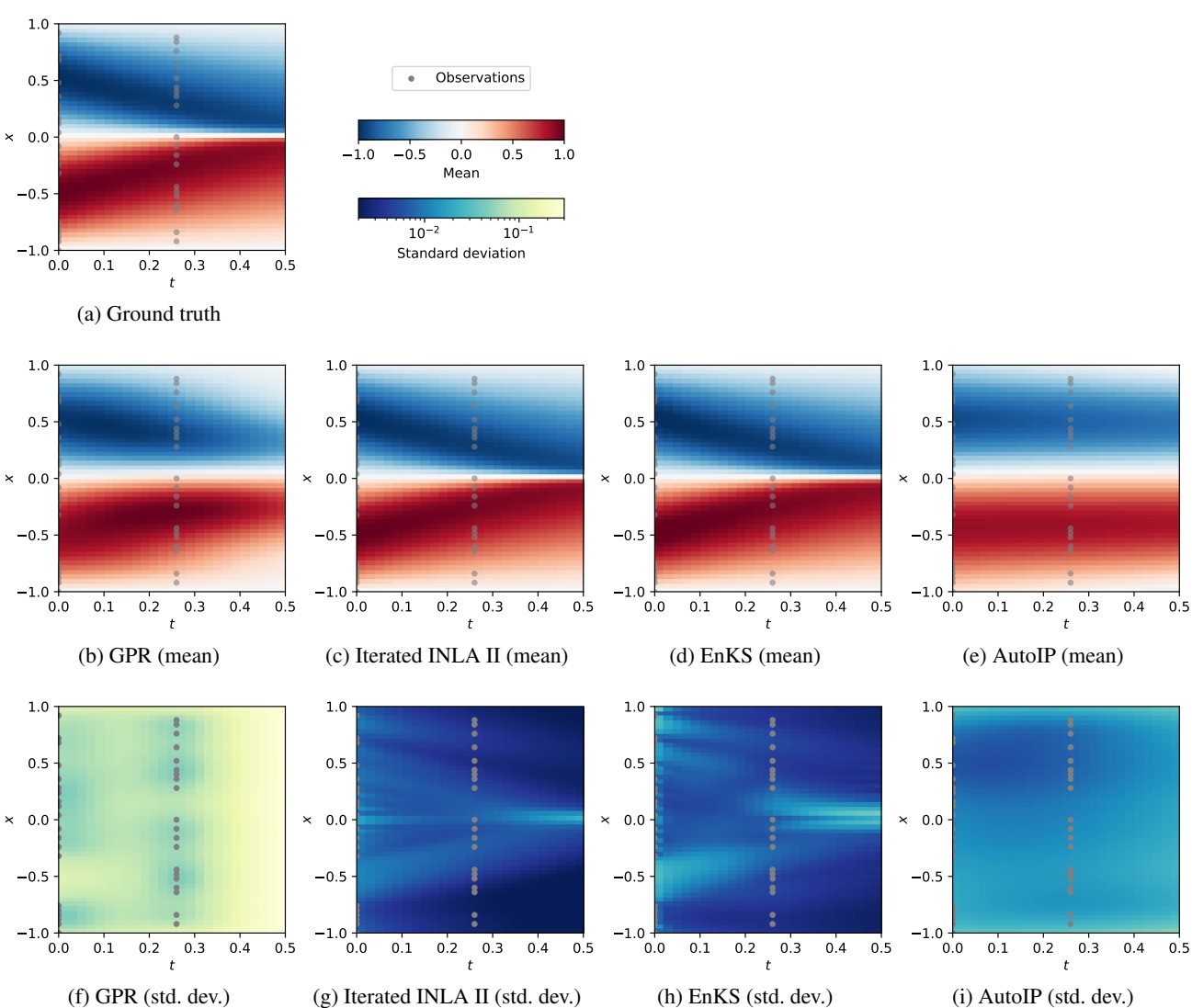

(a) Ground truth

(b) GPR (mean)  (c) Iterated INLA II (mean)  (d) EnKS (mean)  (e) AutoIP (mean)

(f) GPR (std. dev.)  (g) Iterated INLA II (std. dev.)  (h) EnKS (std. dev.)  (i) AutoIP (std. dev.)

Figure 5: Results on the Burgers' experiment

## E.2 ALLEN-CAHN EXPERIMENT

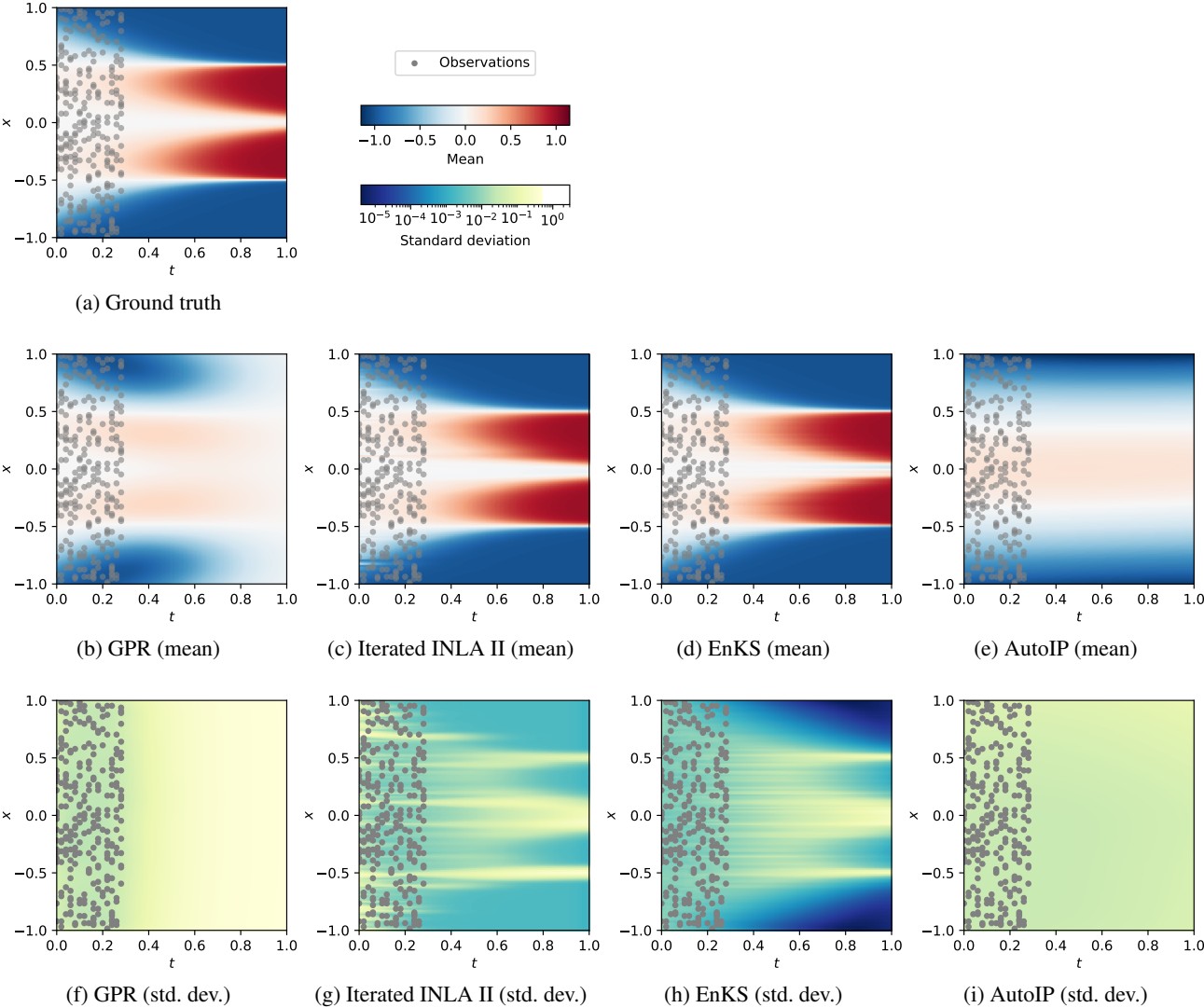

Figure 6: Results on the Allen-Cahn experiment

## E.3 KORTEWEG-DE VRIES EXPERIMENT

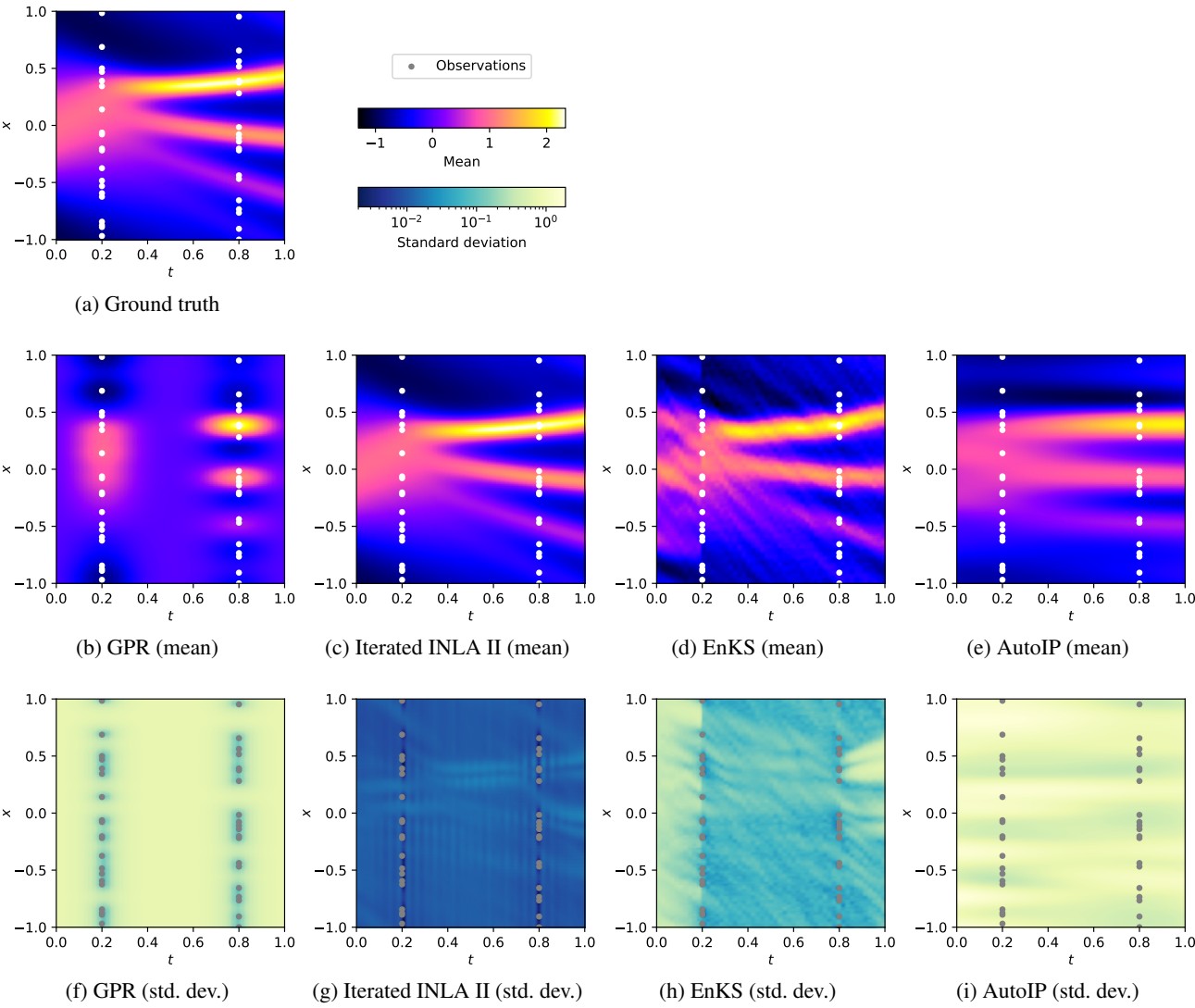

Figure 7: Results on the KdV experiment