# OpenReview forum: "Iterated INLA for State and Parameter Estimation in Nonlinear Dynamical Systems"
_auai.org/UAI/2024/Conference — UAI 2024 poster_

### Official Review · Reviewer_TvD7 · 2024-03-16

**Q2-1 Originality-Novelty:** 3
**Q2-2 Correctness-Technical Quality:** 3
**Q2-5 Clarity Of Writing:** 3

**Q1 Summary And Contributions:**

We extend their method to handle nonlinear PDE priors by iteratively linearizing the PDE to produce GP approximations to the prior at each iteration, and subsequently using INLA to update the state and parameter estimates, and subsequently using INLA to update the state and parameter estimates.

**Q2-3 Extent To Which Claims Are Supported By Evidence:**

2: Fair: the main claims are somewhat supported by evidence (but the experimental evaluation may be weak, or does not match entirely with the claims, important baselines may be missing, proofs contain important ideas but lack rigor, algorithmic details are only discussed superficially, references are imprecise, assumptions are not sufficiently motivated or explicated, etc.).

**Q2-4 Reproducibility:**

2: Fair: key resources (e.g. proofs, code, data) are unavailable but key details (e.g. proof sketches, experimental setup) are sufficiently well-described for an expert to confidently reproduce the main results.

**Q3 Main Strengths:**

This paper proposes a novel data assimilation algorithm based on the INLA methodology that effectively learns the state and the parameters in nonlinear dynamical systems from data that is sparsely distributed.
The paper was well organized. Overall, the paper is sound, and the results are presented in an appropriate way. Tables are illustrative. The method was tested on various datasets.

**Q4 Main Weakness:**

Some sub problems like  non-Gaussian likelihoods, Markovian structure, are not addressed.

**Q5 Detailed Comments To The Authors:**

linearising  --> linearising , last paragraph in introduction section

**Q9 Complying With Reviewing Instructions:**

Yes

---

> ### Author Rebuttal · Authors · 2024-04-06
>
> We thank the reviewer for your feedback and appreciate that the reviewer found our paper well organised with sound results.
>
> For updates on our experiments, please find our response to Reviewer NqhQ.
>
> Regarding extensions to non-Gaussian likelihood and exploiting Markovianity, we opted to defer investigating this to future work as we believe these are interesting extensions on their own right. This also prevents us from overloading our paper with content so as to not eclipse the main ideas of the present work. Below, we elaborate on how we plan on approaching these extensions.
>
> For non-Gaussian likelihoods, INLA originally handles this by considering nested Laplace approximations, which linearises the likelihood around the MAP estimate of the state. We believe that this can be incorporated naturally into our algorithm by sequentially linearising our likelihood and the prior model, in a fashion similar to approximate expectation propagation [1].
>
> Incorporating the Markovian structure within INLA is trickier, since Markovian approaches typically rely on propagating the dense covariance matrix forward in time, and not the sparse precision, as required by INLA. An idea here might be to combine ensemble Kalman methods with INLA, where the state estimates are obtained by EnKS and the parameter estimates are obtained in a fashion akin to INLA. The latter requires the computation of the cost (46), which involve the precision matrix, making it expensive if we only have access to the covariance. However, the covariance matrix obtained by EnKS are always very low-rank, which we may use to our advantage to speed up computations. This sort of computational trick is employed for example in the ensemble square-root filter [2].
>
> Another interesting extension is to consider a parallelised version of INLA as proposed in [3] to make use of modern multicore compute hardwares such as GPUs to further accelerate computations.
>
> Indeed, there are many other extensions we can consider; we believe that our work will serve as an important stepping stone that can integrate the INLA methodology within nonlinear data assimilation that we hope will open doors for new possibilities in uncertainty quantification of states and parameters for moderate-to-high dimensional systems.
>
> __References__
>
> [1] Iterative State Estimation in Non-linear Dynamical Systems Using Approximate Expectation Propagation. Kamthe et al., 2022.
>
> [2] Parallelized integrated nested Laplace approximations for fast Bayesian inference. Gaedke-Merzh \"auser et al., 2022.
>
> [3] Ensemble Square Root Filters. Tippett et al., 2003.

---

### Official Review · Reviewer_BJ8T · 2024-03-22

**Q2-1 Originality-Novelty:** 3
**Q2-2 Correctness-Technical Quality:** 3
**Q2-5 Clarity Of Writing:** 3

**Q1 Summary And Contributions:**

The paper introduces iterated integrated nested Laplace approximation, for inferring the parameters and state of a non-linear dynamical system.

**Q2-3 Extent To Which Claims Are Supported By Evidence:**

2: Fair: the main claims are somewhat supported by evidence (but the experimental evaluation may be weak, or does not match entirely with the claims, important baselines may be missing, proofs contain important ideas but lack rigor, algorithmic details are only discussed superficially, references are imprecise, assumptions are not sufficiently motivated or explicated, etc.).

**Q2-4 Reproducibility:**

2: Fair: key resources (e.g. proofs, code, data) are unavailable but key details (e.g. proof sketches, experimental setup) are sufficiently well-described for an expert to confidently reproduce the main results.

**Q3 Main Strengths:**

* The method has good results in some of the experiments.
* Good explanation of the problem and background.

**Q4 Main Weakness:**

* The experiments are only on toy problems and simulated benchmarks.
* No source code is available.
* There is no mention of the computational efficiency of the method.
* I found the explanation of the method's technical details hard to follow.

**Q5 Detailed Comments To The Authors:**

* eq (2): what is the $\mathcal{B}$ doing in this notation? here, shouldn't this just be u(x,y)=0 for x in boundary(D)?
 * "Gauß-Newton"
   the spelling "Gauss" is much more common
 * Figure 4: what are the grey dots?
 * Algorithm 1: line 4: What is "$\mathcal{L(θ)}$"? Everywhere else $\mathcal{L}$ does not have a parameter θ.
 * Eq (23) and Algorithm 1 line 5: the sign of the residual is opposite of what I would expect, why is it not r = L[u] - L_0·u?

**Q9 Complying With Reviewing Instructions:**

Yes

---

> ### Author Rebuttal · Authors · 2024-04-06
>
> We thank the reviewer for taking the time to go through our work carefully. Please find our responses to each of your questions below.
>
> > The experiments are only on toy problems and simulated benchmarks
>
> Indeed, this is a limitation of our work, being only proof-of-concept. We plan to investigate this in future work by first extending our approach to deal with multiple output components and also to non-Gaussian likelihoods.
>
> > No source code is available.
>
> We have not included our code in order to retain our anonymity. However, we are planning on making our code open source and include a link to it in the paper for the camera-ready version.
>
> > Technical details hard to follow.
>
> While we aimed to make our work more accessible to the wider audience, we did struggle to explain certain aspects of our method without delving into some technicalities. If there are parts in particular that the reviewer believes should be explained more clearly, we would be happy to improve it.
>
> > eq(2): what is the $\mathcal{B}$ doing in this notation?
>
> The notation $\mathcal{B}$ used here is to denote arbitrary boundary conditions, not limited to the Dirichlet conditions (for example, periodic, Neumann and Robin conditions). We will make this clear in the text.
>
> > "Gau\ss-Newton" the spelling "Gauss" is more common.
>
> We will correct the spelling in a subsequent version of our manuscript.
>
> > What are the grey dots in Figure 4?
>
> These denote noisy observations of the underlying field. We will indicate this in the caption.
>
> > What is "$\mathcal{L}(\theta)$"? Everywhere else $\mathcal{L}$ does not have a parameter $\theta$.
>
> The operator $\mathcal{L}$ is assumed to depend on the parameter $\theta$ all the time. However for the most part, we have opted to exclude the $\theta$ dependence from our notation to prevent overloading with symbols. As the reviewer suggest, it is indeed unnatural to have the $\theta$ dependence appear only in Alg. 1. Hence, to keep our notations consistent, we will change $\mathcal{L}(\theta)$ in Alg. 1 to $\mathcal{L}$.
>
> > The sign of the residual is the opposite of what I would expect.
>
> We have checked that the sign here is correct. To go from equation (25) to (26), the sign flips since $\mathcal{L}$ is proportional to $-\mathcal{M}$ and not $\mathcal{M}$, due to how we defined the $\mathcal{L}$ operator (see (18)).

---

### Official Review · Reviewer_NqhQ · 2024-03-23

**Q2-1 Originality-Novelty:** 3
**Q2-2 Correctness-Technical Quality:** 3
**Q2-5 Clarity Of Writing:** 3

**Q10 Ethical Concerns:**

No.

**Q1 Summary And Contributions:**

This paper presents a novel Integrated Nested Laplace Approximation (INLA) application for nonlinear dynamical system predictions. The authors utilize Gaussian Model Random Fields and formulate the learning problem into a Bayesian framework. It discusses cases when Iterated INLA with and without knowing the parameters. The numerical studies are supporting this new method as it generally outperforms baseline methods (including GPR and the ensemble Kalman filter).

**Q2-3 Extent To Which Claims Are Supported By Evidence:**

3: Good: the main claims are supported by convincing evidence (in the form of adequate experimental evaluation, proofs, (pseudo-)code, references, assumptions).

**Q2-4 Reproducibility:**

3: Good: key resources (e.g. proofs, code, data) are available and key details (e.g. proofs, experimental setup) are sufficiently well-described for competent researchers to confidently reproduce the main results.

**Q3 Main Strengths:**

This paper is technically strong and well-written. It formulates the mathematical problem very clearly and provides related guarantees for nonlinear systems. The background discussion is precise and clean. In general, this is a very good paper.

INLA itself is a very powerful tool that enables massive improvement in computation for many types of problems. However, a huge drawback is that the Laplace approximation normally has to follow a "Gaussian assumption". Therefore applying this to nonlinear PDEs is challenging, and, from the experimental results, the iINLA method achieves very good performances compared to baseline methods.

**Q4 Main Weakness:**

The reviewer would still like to point out some weaknesses of this paper.

1. Even if the problem formulation, mathematical definition, algorithm, and experiments (including the Appendix) are very well-written, to the reviewer, the introduction is somehow harder to read. Please refer to Q5 (detailed comments for more details).

2. The authors are comparing iINLA to ensemble Kalman filter and Gaussian process. However, it is crucial to investigate whether other more recent studies have addressed this problem before. The reviewer will be surprised if ensemble KF is the most recent literature (baseline method) to look at.

**Q5 Detailed Comments To The Authors:**

1. The writing of introduction is a bit problematic. In general, it is using too many acronyms which assumes all audiences are coming form the same background. For example, GPR is only introduced in the appendix but is pretty heavily used throughout the entire manuscript. Additionally, it might probably be a good idea to introduce stochastic PDEs before writing SPDEs.

1.1 May the authors spend a few more lines to explain the challenge of this problem?

2. What is the computational cost of iINLA compared to other methods?

3. It appears in Figure 3 that the ensemble KF method is more biased compared to the iINLA. Could the authors probably comment on the reason of this phenomenon? The reviewer thinks this bias might be natural for the enKF method and the bias will converge statistically (with more samples, less noise, ...).

4. The reviewer likes the PDE results.

**Q9 Complying With Reviewing Instructions:**

Yes

---

> ### Author Rebuttal · Authors · 2024-04-06
>
> We thank the reviewer's time for going through our work thoroughly. To address the reviewer's comments about the baseline comparisons, our adoption of the EnKS as a primary baseline is justified by reasons which we describe below.
>
> Firstly, we distinguish between an empirical and a fully Bayes approach -- the former only provides point estimates of the model parameters, whereas the latter provides uncertainty quantification (UQ) on both the state and parameters. In the machine learning literature, there are many new developments in the former setting (e.g. AutoIP, latent force models, latent SDE based methods, etc...), however, in this paper, we are primarily interested in the latter. In short, there aren't many viable methods outside of expensive MCMC methods that currently tackle the fully Bayesian setting with the exception of state-augmented EnKS.
>
> Specifically, for the fully Bayesian approach, most recent works that we are aware of consider Particle MCMC (PMCMC) methods, which uses the particle filter to get state estimates and MCMC to get the parameter estimates (the PMMH algorithm we use in our pendulum experiment is one example of this). However, this can be tricky to deploy in practice as it requires a large number of particles to get accurate solutions, leading to high computational costs. Furthermore, in moderately high dimensions, these methods are prone to mode collapse. Recent developments in the field have begun to address the issue of mode collapse on toy problems with some success (see [1] for an excellent review on this topic). Despite this, the use of state-of-the-art PMCMC methods remain relatively obscure in practice as there is still no general recipe to construct proposals that prevent mode collapse [2], together with its high computational cost, and importantly, lack of open source code. Hence, the state-augmented EnKS remains a popular method in data assimilation, due to its conceptual simplicity, ease of implementation and its surprising effectiveness, making it an important baseline to compare against.
>
> Regarding EnKS-type state and parameter methods, we have found more recent work on the so-called _Iterated EnKS_ (hereafter, _iEnKS_ ) to improve state and parameter estimation [3]. We have now added new comparisons with this baseline with the following results:
>
> __Stochastic Pendulum:__
>
> |          | RMSE | MNLL |
> | ------ | ------- | ------- |
> | EnKS | 0.176 +/- 0.012 | -0.495 +/- 0.084 |
> | *iEnKS* | 0.227 +/- 0.072 | 6.081 +/- 4.793 |
> | iINLA-II | **0.175 +/- 0.014** | **-0.667 +/- 0.056** |
>
> __Burgers:__
>
> |          | RMSE | MNLL |
> | ------ | ------- | ------- |
> | EnKS | 0.008 +/- 0.001 | -3.67 +/- 0.11 |
> | *iEnKS* | **0.006 +/- 0.001** | **-3.97 +/- 0.05** |
> | iINLA-II | 0.009 +/- 0.001 | -3.49 +/- 0.34 |
>
> __Allen-Cahn:__
>
> |          | RMSE | MNLL |
> | ------ | ------- | ------- |
> | EnKS | **0.028 +/- 0.001** | **-4.08 +/- 0.076** |
> | *iEnKS* | 0.227 +/- 0.072 | 6.081 +/- 4.793 |
> | iINLA-II | 0.009 +/- 0.001 | -3.49 +/- 0.34 |
>
> __Korteweg-de-Vries:__
>
> |          | RMSE | MNLL |
> | ------ | ------- | ------- |
> | EnKS | 0.228 +/- 0.029 | -0.010 +/- 0.263 |
> | *iEnKS* | 0.131 +/- 0.021 | 2807 +/- 650 |
> | iINLA-II | **0.010 +/- 0.000** | **-3.28 +/- 0.04** |
>
> Overall, we see that with the exception of the Burgers' example, the iterated EnKS did not lead to improvements over the vanilla EnKS.
>
> To address the remaining comments by the reviewer:
>
> > The writing of introduction is a bit problematic.
>
> We agree and will use fewer acronyms in the introduction.
>
> > May the authors spend a few more lines to explain the challenge of this problem?
>
> The main challenge is accurate state and parameter estimation with good uncertainty quantification without reliance on expensive MCMC. Our goal is to propose an alternative methodology based on INLA, which has originally been proposed in spatiotemporal statistics as a faster alternative to MCMC. Thus, we are interested in seeing whether the use of INLA is also viable in nonlinear data assimilation.
>
> > What is the computational cost of iINLA compared to other methods?
>
> Please find our response to reviewer SYfj.
>
> > It appears in Figure 3 that the ensemble KF method is more biased compared to the
> iINLA.
>
> This behaviour is expected, since EnKS is known to not be able to learn stochastic parameters accurately even under large sample limits (see [4] for a detailed discussion).
>
> __References__
>
> [1] Particle filters for high-dimensional geoscience applications: A review. van Leeuwen et al., 2019.
>
> [2] An Introduction to Sequential
> Monte Carlo. Chopin and Pappaspiliopoulos, 2020.
>
> [3] Joint state and parameter estimation with an iterative ensemble Kalman smoother. Bocquet and Sakov, 2013.
>
> [4] State and parameter estimation in stochastic dynamical models. DelSole and Yang, 2010.

---

### Official Review · Reviewer_SYfj · 2024-03-23

**Q2-1 Originality-Novelty:** 3
**Q2-2 Correctness-Technical Quality:** 4
**Q2-5 Clarity Of Writing:** 4

**Q1 Summary And Contributions:**

This paper addresses the interesting problem of data assimilation when the dynamical model takes the form of a nonlinear stochastic PDE. The authors present iINLA, or iterated INLA, which is needed because the resulting posterior state model is non-Gaussian. Their approach is simple: at each iteration, the nonlinear model is linearized about a fixed point and approximated by a Gaussian MRF, which facilitates use of INLA for marginal posterior state updates and parameter estimation.  iINLA is similar to INLA-SPDE but the nonlinearity in INLA-SPDE appears in the observed operator that links the latent states to the observations rather than directly on the latent SPDE describing the states. The relation of iINLA to 4D-Variational data assimilation is discussed and the new method is compared to 3 other methods on simulations from 4 benchmark nonlinear dynamical systems.

**Q2-3 Extent To Which Claims Are Supported By Evidence:**

3: Good: the main claims are supported by convincing evidence (in the form of adequate experimental evaluation, proofs, (pseudo-)code, references, assumptions).

**Q2-4 Reproducibility:**

3: Good: key resources (e.g. proofs, code, data) are available and key details (e.g. proofs, experimental setup) are sufficiently well-described for competent researchers to confidently reproduce the main results.

**Q3 Main Strengths:**

Main strengths:  The paper is generally well-written and follows a natural progression of ideas from linear to nonlinear dynamic systems. While the idea of using linear approximations at each iteration is not novel, applying linearization on the nonlinear operator of the stochastic model, and working out the corresponding technical details, is. Two algorithms are presented, one for when the model parameters are known, but the latent states are unknown, and another when both the model parameters and latent states are unknown. Simulation results show that iINLA-II, when both parameters and states are unknown, is competitive with or better than ensemble Kalman filter/smoother on the benchmark systems tested.  Proofs for theoretical results are provided in the supplementary material, as are more details results from the simulations. The technical details seem to be correct, though there are some results in the supplementary material that I did not verify in great detail.

**Q4 Main Weakness:**

Main weaknesses: While I appreciate the details and the intuition provided in the main manuscript and the supplementary material, this paper (inclusive of the supplementary material) is very long, sitting at 26 pages.  The time reduction needed for inference via iINLA compared to stochastic MC is provided in one single run of the alternating pendulum system, but this does not give an idea of how computation time compares to other methods or on the more complicated systems.  One single run hardly provides useful information – averaging over the runs would have been better. Finally, iINLA was only demonstrated on toy examples.  The authors do comment on this as a limitation. One can only hope that it works as well and scales appropriately on realworld applications but this was not demonstrated in the paper.

**Q5 Detailed Comments To The Authors:**

1. Please provide more details on run times across methods, especially EnsKS since the two seem to be more or less competitive.
2. It seems that all equations are numbered, but not all are referenced by number in the manuscript.  I would only number those equations to which you refer in the main text.
3. There is no explicit Relation to Other Work section. Instead, these other works are discussed in the introduction and in Remark 3.1.  The authors may consider pulling these methods together formally in a labelled Relation to Other Work paragraph at the end of the Introduction, just before the Section 2. The label need not be a section or subsection header per se, but just bolded in text before the discussion of other work begins.

**Q9 Complying With Reviewing Instructions:**

Yes

---

> ### Author Rebuttal · Authors · 2024-04-06
>
> We thank the reviewer for providing us with valuable feedback on our manuscript.
>
> To address the point about runtimes, firstly, the computational complexity of Sequential Monte Carlo (more specifically, the particle marginal Metropolis Hastings (PMMH) algorithm), iterated INLA and the ensemble Kalman smoother are as follows:
>
> - PMMH: $\mathcal{O}(N_p S)$, where $N_p$ is the number of particles in the parameter space and $S$ is the complexity to run a particle smoother, which, for the bootstrap filter has complexity $\mathcal{O}(N_s)$, where $N_s$ is the number of particles in the state space. This is slow since typically a large number of particles are required for the Metropolis-Hastings algorithm to converge.
>
> - Iterated INLA: $\mathcal{O}(N_i I)$, where $N_i$ is the number of iterations and $I$ is the complexity of one iteration of INLA (see Section A.1 of our paper for more details on this). There is no significant difference between type I and II updates.
>
> - EnKS: $\mathcal{O}(TN_e^3 + TN_e^2 N_y + TN_e N_d)$ for the deterministic variant, where $T$ is the number of time steps, $N_e$ is the ensemble size, $N_y$ is the observation dimension at a single time step, and $N_d$ is the state dimension at a single time step. This is fast when using small ensemble sizes.
>
> On the specific example of the nonlinear pendulum that we used in our experiments (using our updated code), the concrete run times were as follows (mean and standard deviation across five different seeds):
>
> - PMMH (10,000 samples): $1541 \pm 38$s
>
> - iINLA (25 iterations): $67.261 \pm 0.939$s
>
> - EnKS (100 ensembles): $3.071 \pm 0.314$s
>
> We see that the run time of iterated INLA sits between EnKS and SMC. To justify the additional computational cost of iINLA over EnKS, we recall the following performance gains, which we discuss in the paper: (1) more accurate estimate of the stochastic parameters, (2) similar or better state estimates compared to EnKS, (3) increased robustness to numerical discretisation. Ultimately, it is up to the user and the task to determine whether these benefits outweigh the use of EnKS to obtain the state and parameter estimates quickly. We also note that our code is only proof-of-concept and relies on our own implementation of INLA that is not fully optimised -- integrating with the R-INLA package will likely lead to further speedup than what we observe above.

---

### Meta-Review · Area_Chair_4vix · 2024-04-15

The reviewers agree that this paper presents an interesting problem of data assimilation  based on iteratively linearising the dynamical model. The paper is well written and the authors have convincingly replied to their comments